# Sustained IFN signaling is associated with delayed development of SARS-CoV-2-specific immunity

Plasma RNAemia, delayed antibody responses and inflammation predict COVID-19 outcomes, but the mechanisms underlying these immunovirological patterns are poorly understood. We profile 782 longitudinal plasma samples from 318 hospitalized patients with COVID-19. Integrated analysis using k-means reveals four patient clusters in a discovery cohort: mechanically ventilated critically-ill cases are subdivided into good prognosis and high-fatality clusters (reproduced in a validation cohort), while non-critical survivors segregate into high and low early antibody responders. Only the high-fatality cluster is enriched for transcriptomic signatures associated with COVID-19 severity, and each cluster has distinct RBD-specific antibody elicitation kinetics. Both critical and non-critical clusters with delayed antibody responses exhibit sustained IFN signatures, which negatively correlate with contemporaneous RBD-specific IgG levels and absolute SARS-CoV-2-specific B and CD4+ T cell frequencies. These data suggest that the "Interferon paradox" previously described in murine LCMV models is operative in COVID-19, with excessive IFN signaling delaying development of adaptive virus-specific immunity.

Coronavirus disease 2019 (COVID-19) caused by severe acute respiratory syndrome coronavirus 2 (SARS-CoV-2) infection is a heterogeneous disease that ranges from asymptomatic infection to fatal outcome. Qualitative and kinetic differences in viral loads and immune responses have been associated with clinical severity: we[1] and others[2–5] have shown that SARS-CoV-2 plasma viral RNA (vRNA) levels predict fatality in patients with COVID-19. However, some patients succumbed to their infection in the absence of high plasma vRNA, while other individuals with high vRNA survived[1], indicating a role for additional factors.

High amounts of inflammatory cytokines have also been linked with fatal outcome[6–8]. These cytokines are implicated in immuno-pathology since immunomodulatory treatments such as IL-6R antagonists[9,10], systemic corticosteroids[11], and Janus kinase inhibitors[4] improve COVID-19+ patients' survival. Despite the well-established roles of interferon (IFN) pathways in priming of antiviral immunity and evidence that pre-existing defects of type I IFN responses are associated with adverse prognosis[12,13], recombinant type I IFN therapy failed to improve COVID-19 outcomes[14] and may even be detrimental in severe disease[15]. This apparent paradox is consistent with observations that sustained IFN levels impair lung healing[16], while their impact on adaptive immunity remains to be determined. In addition, delayed antibody responses[17,18], possibly linked to disrupted coordination between virus-specific T and B cells[19], have been observed in patients with fatal disease.

Given the highly dynamic nature of COVID-19, binning patients based on clinical characteristics across the duration of their hospitalization can blur our understanding of the disease course. Although several studies have shown outcome associations with immunovirological feature[7,8,20], we still lack a global understanding of the immunovirological kinetics associated with disease heterogeneity. Endotypes, in which patients are grouped based on molecular rather than clinical characteristics, allow a more accurate identification of high-risk patient subsets[21]. The interplay between these molecular

✉ e-mail: michael.hultstrom@mcb.uu.se; guy.wolf@umontreal.ca; daniel.kaufmann@chuv.ch

signatures can be visualized and explored through dimensionality reduction. Potential of Heat-diffusion for Affinity-based Trajectory Embedding (PHATE) is a manifold learning algorithm that computes a nonlinear transformation of the data to represent the latent structure of a dataset in low dimensions[22]. In parallel, a k-means algorithm can be used to group patients into defined clusters sharing similar features.

Using this type of integrative approach on cross-sectional measurements of plasma vRNA, Spike Receptor Binding Domain (RBD)-specific antibody responses, and plasma levels of cytokines and tissue damage markers, we identified four patient clusters corresponding to different systemic responses to acute SARS-CoV-2 infection. These endotypes closely associated with clinical severity. Longitudinal profiling and computational modeling of the antibody responses showed that delayed RBD-specific antibody response was a central feature in two clusters. Using whole-blood transcriptional profiling, we show that patients with this delay have sustained IFN signatures. These signatures were also negatively associated with SARS-CoV-2-specific CD4+ T cell and B cell responses, but not CD8+ T cell responses. These results highlight a role for excessive IFN signaling in disrupting adaptive humoral and cellular immune responses to a human viral infection.

## Results
### Study design, patient characteristics and classification
We investigated prospectively enrolled hospitalized COVID-19+ patients with symptomatic infection and a positive SARS-CoV-2 nasopharyngeal swab (NSW) PCR from two hospitals in Montreal, Quebec, Canada ($n = 242$, discovery cohort) and one in Uppsala, Sweden ($n = 76$, validation cohort). Blood draws were serially done throughout their stay in-hospital at enrollment (day 0) and at 2, 7, 14, and 30 days. Only acute samples, defined as those collected within 30 days of symptom onset ($n = 630$ in the discovery cohort; 152 in the validation cohort), were considered. We previously observed that plasma vRNA, RBD-specific IgG antibodies, cytokines (TNFα, CXCL13, IL-6, IL-23, CXCL8, CCL2 and IL1Ra), and tissue damage markers [Receptor for Advanced Glycation Endproducts (RAGE), Angiopoietin-2 (Ang-2) and surfactant protein D (SP-D)] were associated to fatal outcome when measured 11 days after symptom onset (DSO11)[1], while associations with RBD-specific IgM and IgA levels did not reach statistical significance[1]. For unsupervised data characterization, we considered these 14 measurements in cross-sectional samples taken at DSO11 ($+/- 4$ days, $n = 242$) (Fig. 1A). We first visualized samples on a 2D scatter plot using the PHATE dimensionality reduction algorithm[22]. In parallel, we performed a k-means clustering on the same data. We chose a cluster count of four, as running k-means with a higher number of clusters led to over-fragmentation and small clusters ($n < 20$), preventing adequate subgroup characterization. The clustering resulted in two smaller clusters (1: $n = 38$ and 2: $n = 49$) and two larger ones (3: $n = 73$ and 4: $n = 82$), which strongly aligned with regions of the PHATE embedding (Fig. 1B). No clinical or demographic data was used for computing the PHATE embedding and clustering.

### Hospitalized patients display four distinct endotypes of early plasma immunovirological profiles following SARS-CoV-2 infection
We examined how the parameters used to create the PHATE embedding differed between patient clusters at DSO11. Nearly all patients in cluster 1 had detectable vRNA and at higher amounts than the other clusters, with cluster 3 having the second-highest levels (Fig. 1C). As previously described[1], we observed a strong association between cytokines and most of the tissue damage markers across our cohort (Fig S1A). To integrate the overall quantities of cytokines, we created a cytokine score through the linear combination of the 7 cytokines surveyed (see "Methods" for details). This score followed a stepwise decrease between clusters 1, 2, 3, and 4 (Fig. 1D). A second score created with the three markers of tissue damage (TD score) also showed a

stepwise decrease from cluster 1 to 4, although there were no statistically significant differences between clusters 3 and 4 (Fig. 1E). Both scores were strongly correlated with one another (Fig. S1B), in line with the association between tissue damage and inflammation. At this DSO11 timepoint, the RBD-specific IgG response was high in clusters 2 and 4, low in cluster 1, and undetectable in most participants of cluster 3 (Fig. 1F), with analogous patterns observed for RBD-specific IgM (Fig. S1C) and IgA (Fig. S1D).

To assess how these different immunovirological patterns associated with disease severity, we examined the contemporaneous patient status based on the level of respiratory support received (Moderate = no supplemental oxygen; Severe = oxygen on nasal cannula; Critical = non-invasive or invasive mechanical ventilation). Clusters 1 and 2 were enriched for critical patients, while clusters 3 and 4 mainly contained non-critical patients (Figs. 1G, S1E). Cluster 1 identified the most severe cases, as reflected by outcome: 50% of patients in cluster 1 died within 60 days of symptom onset, while fatal outcome was observed in a minority of the other three clusters (Figs. 1H, S1F). This was also reflected in the duration of hospitalization (S1G). Age distribution was similar between clusters (Table 1, Fig. S1H) and across the embedding (Fig. S1I). A similar observation was made for sex (Fig. S1JKL). The distributions of ethnicities were comparable across the four cluster (Fig. S1M). Other demographics were similar between cohorts, except for the enrichment of pre-existing renal failure in cluster 1 (Table 1).

Our analytical approach therefore identified four immunovirological endotypes in SARS-CoV-2 infection at DSO11 (Fig. 1I) that not only aligned with contemporaneous disease severity but also delineated probability of survival among critical cases.

### Replication of the high fatality cluster 1 in an external validation cohort
The distinct validation cohort, recruited in a Swedish hospital, differed from clusters 1 and 2 of the discovery cohort for age, sex, and prevalence of some pre-existing conditions (Table 2). The validation cohort had a greater incidence of mechanical respiratory support, in line with recruitment of this cohort exclusively from the ICU (Table 2). Despite these differences, the incidence of fatal outcome was similar. The PHATE and k-mean analyses were performed using a subset of measurements common between both the discovery and validation cohorts, including the 3 antibody measurements, plasma vRNA, 5 of 7 cytokines, and 1 of 3 tissue damage markers (see "Methods" for details). PHATE produced two natural clusters that again aligned with k-means analysis (Clusters V1 and V2) (Fig. 1J). The two clusters recapitulated the immunovirological patterns identified in clusters 1 and 2 in the discovery dataset. Cluster V1 showed higher viral load (Fig. 1K), inflammation (Fig. 1L), and tissue damage (Fig. 1M). The differences in vRNA and tissue damage markers were less pronounced, likely due to differences in measurement methods and the use of a subset of analytes compared to the discovery cohort. Cluster V1 also had lower N-specific antibodies at DSO11 (Fig. 1N) than cluster V2. Nonetheless, cluster V1 was strongly enriched in fatal outcome (Fig. 1O). The strong reproducibility between both cohorts, despite differences in some clinical parameters between cohorts, the use of a subset of the original set of analytes and differences in the SARS-CoV-2 target of the antibodies measured, validates the use of the PHATE/k-means analysis of early plasma profile to classify patient heterogeneity.

### Delayed antibody kinetics is associated with protracted plasma vRNA over a wide range of disease severity
Given the differences in antibody levels among the discovery clusters at DSO11, we examined whether these resulted from either a delay or an inability in generating anti-RBD antibodies. We compared antibody levels at a later timepoint (DSO20 $+/- 4$ days) and saw no significant difference between patient clusters (Fig S2A), indicating a late, but

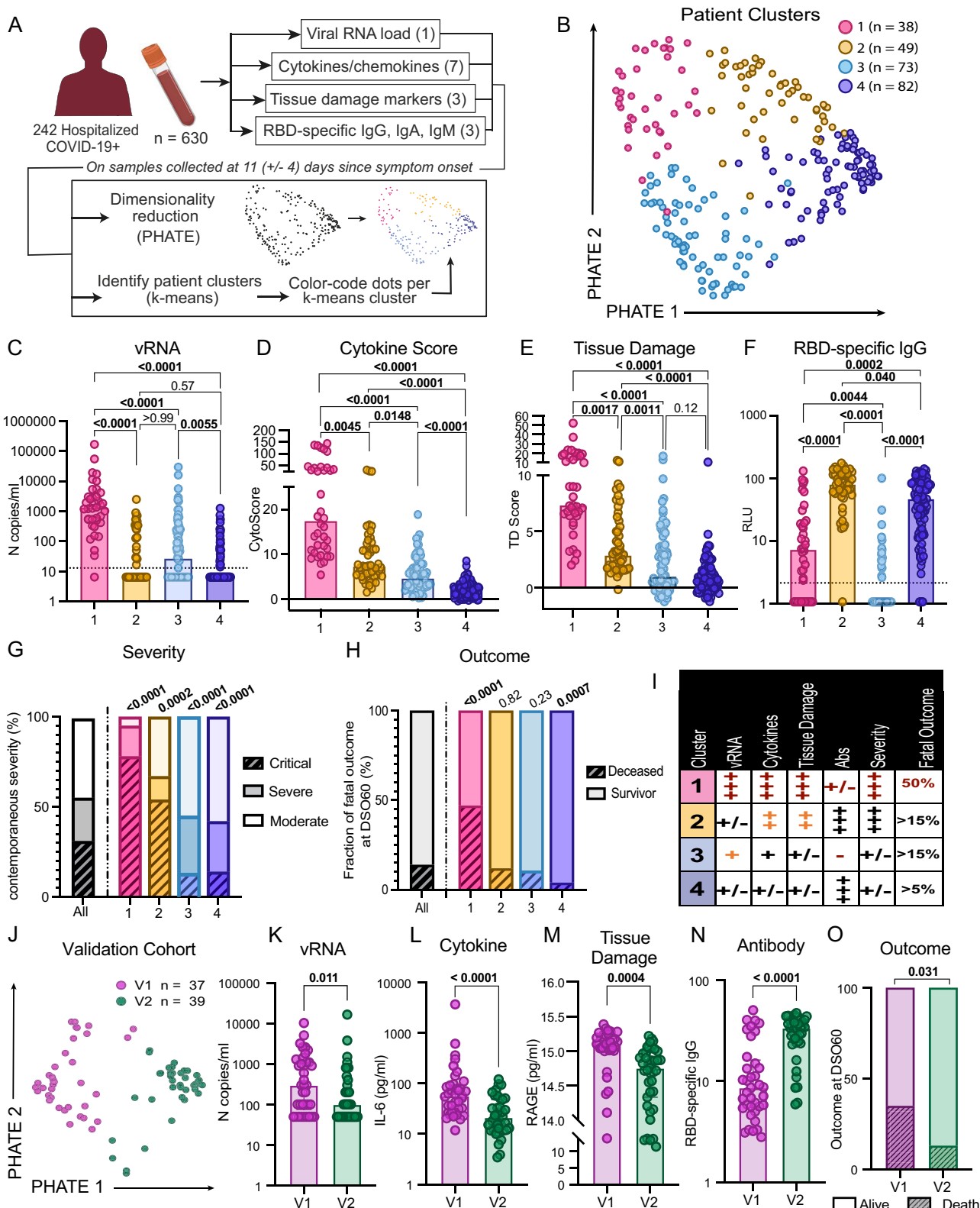

ultimately comparable, response. To compute the antibody kinetics, we combined the RBD-specific antibody measurements of all samples within a cluster and modeled a logistic curve on the quantity of antibodies per patient cluster per day (Fig. 2A). Statistical comparison using bootstrap analysis (see "Methods" for details) revealed that patient clusters had significantly distinct timings in the generation of RBD-specific IgG responses. Both clusters 2 and 4 reached 50% of maximum RBD-specific IgG amount ($DSO_{50\%}$) before DSO11 (Fig. 2A),

which is why they had already high levels of antibodies at DSO11. Conversely, cluster 1 reached $DSO_{50\%}$ around DSO13, and cluster 3 was latest, with a $DSO_{50\%}$ reached at 17 days. This delayed antibody response was also observed in the validation cohort (Fig. S2BC). We also measured the evolution of RBD-specific IgM (Fig. S2D) and IgA (Fig. S2E) and observed delayed responses in Clusters 1 and 3 that were similar to those observed for IgG. Thus, this delayed kinetics cannot be selectively attributed to impaired class switching.

**Fig. 1 | Hospitalized patients display four distinct endotypes of early plasma immunovirological profiles following SARS-CoV-2 infection. A** Study design. Serial blood samples were collected among hospitalized NSW PCR-confirmed COVID-19+ patients. Samples were assessed for plasma viral RNA, seven cytokines, three tissue damage markers, and three SARS-CoV-2 RBD-specific antibody isotypes. On samples collected 11 (+/−4) days after symptom onset (DSO11), all 14 parameters were combined for visualization by PHATE and used to calculate patient clusters by k-means. Patient cluster was then used to color-code PHATE embedding. **B** DSO11 samples identified four patient clusters across 242 hospitalized COVID-19+ patients. **C–F** At DSO11, plasma concentration across four patient clusters of (**C**) viral RNA; (**D**) Cytokine Score obtained from the linear combination of all seven cytokines; (**E**) score of tissue damage obtained from the linear combination of all three markers of tissue damage, and (**F**) SARS-CoV-2 RBD-specific IgG. **G, H** Percentage of the whole cohort or per patient cluster (**G**) with critical

(hashed), severe (saturated), or moderate (faint) disease; (**H**) with fatal outcome (hashed). **I** Summary table of four patient clusters in the discovery cohort. **J** Validation cohort of 76 hospitalized COVID-19+ patients. SARS-CoV-2-specific IgG, IgM and IgA, vRNA and cytokines and tissue damage markers were measured at DSO11. PHATE embedding and k-means clustering were performed as for the discovery cohort. **K–O** Comparison, between validation cluster (V)1 and V2, of plasma levels of (**K**) SARS-CoV-2 vRNA; (**L**) IL-6; (**M**) RAGE or (**N**) N-specific IgG. **O** Outcome at DSO60 (hashed being fatal outcome). N discovery cohort: 1 = 38; 2 = 49; 3 = 73; 4 = 82 (242 in total). N validation cohort: V1 = 37; V2 = 39 (76 in total). **C–F** Kruskal-Wallis with Dunn's multiple comparison tests. Adjusted *p* values are shown. **G, H** Two-sided Fisher's exact test compares the proportion of hashed groups in one cluster versus all others. For (**G**), statistical comparison is between critical and non-critical. **K–O** Two-sided Mann-Whitney tests. Medians are shown in bar charts. Source data are provided as Source_Data_File.xlsx.

## Table 1 | Demographics and characteristics of hospital stay per patient cluster of the Discovery cohort

| Variable | Entries | All | 1 | 2 | 3 | 4 | Stats |
|---|---|---|---|---|---|---|---|
| n | | 242 | 38 | 49 | 73 | 82 | |
| Age | [median (IQR#)] | 66.4 (52.4 – 78.8) | 71.8 (64.4 – 78.8) | 66.9 (58.3 – 79.5) | 64.8 (54.1 – 82.0) | 57.6 (48.2 – 74.3) | **0.032** |
| Sex | | | | | | | |
| | Male | 141 (58%) | 25 (66%) | 27 (55%) | 38 (52%) | 51 (62%) | 0.43 |
| | Female | 101 (42%) | 13 (34%) | 22 (45%) | 35 (48%) | 31 (38%) | |
| Max respiratory Support throughout hospital stay | | | | | | | |
| | No O$_2$ | 86 (36%) | 5 (13%) | 10 (20%) | 36 (49%) | 35 (43%) | **0.0001** |
| | NC | 63 (26%) | 5 (13%) | 8 (16%) | 22 (30%) | 28 (34%) | **0.028** |
| | NIV | 32 (13%) | 7 (18%) | 9 (18%) | 5 (7%) | 11 (13%) | 0.20 |
| | ETI | 59 (24%) | 20 (53%) | 22 (45%) | 10 (14%) | 7 (9%) | **<0.0001** |
| | ECMO | 2 (1%) | 1 (3%) | 0 (9%) | 0 (0%) | 1 (1%) | 0.44 |
| Days of hospitalization [median (IQR)] | | 14 (8.0 – 27.0) | 24 (10.0 – 36.0) | 19 (12 – 41.5) | 14 (7.0 – 25.0) | 9 (5.3 – 15.8) | **<0.0001** |
| ICU admission | | 89 (37%) | 27 (71%) | 29 (59%) | 15 (21%) | 18 (22%) | **<0.0001** |
| Days in ICU [median (IQR)] | | 14 (5.0 – 31.0) | 24 (8.5 – 34.0) | 19 (7.0 – 35.0) | 14 (9.0 – 25.5) | 4.5 (3.3 – 6.8) | **0.0003** |
| Metabolic risk factors | | | | | | | |
| | None | 75 (31%) | 10 (26%) | 14 (29%) | 24 (33%) | 27 (33%) | 0.85 |
| | One or more | 167 (69%) | 28 (74%) | 35 (71%) | 49 (67%) | 55 (67%) | |
| | Obese | 33 (14%) | 5 (13%) | 12 (24%) | 10 (14%) | 6 (7%) | 0.053 |
| | Hypertension | 139 (57%) | 26 (68%) | 30 (61%) | 41 (56%) | 42 (51%) | 0.32 |
| | Dyslipidemia | 57 (24%) | 11 (29%) | 16 (33%) | 14 (19%) | 16 (20%) | 0.22 |
| | Diabetes | 86 (36%) | 15 (39%) | 19 (39%) | 27 (37%) | 25 (30%) | 0.69 |
| Chronic diseases | | | | | | | |
| | None | 99 (41%) | 9 (24%) | 19 (39%) | 32 (44%) | 39 (48%) | 0.089 |
| | One or more | 143 (59%) | 29 (76%) | 30 (61%) | 41 (56%) | 43 (52%) | |
| | Chronic Kidney Disease | 41 (17%) | 13 (34%) | 9 (18%) | 10 (14%) | 9 (11%) | **0.013** |
| | Heart Failure | 51 (21%) | 11 (29%) | 11 (22%) | 15 (21%) | 14 (17%) | 0.52 |
| | Respiratory Disease | 42 (17%) | 12 (32%) | 8 (16%) | 9 (12%) | 13 (16%) | 0.078 |
| | Liver Disease | 21 (9%) | 7 (18%) | 5 (10%) | 6 (8%) | 3 (4%) | 0.63 |
| | Immunosuppressed | 20 (8%) | 4 (11%) | 6 (12%) | 6 (8%) | 4 (5%) | 0.47 |
| | Malignancy | 35 (14%) | 5 (13%) | 7 (14%) | 10 (14%) | 13 (16%) | 0.98 |
| | HIV | 5 (2%) | 2 (5%) | 1 (2%) | 1 (1%) | 1 (1%) | 0.50 |
| | Neurological disorder | 30 (12%) | 3 (8%) | 8 (16%) | 8 (11%) | 11 (13%) | 0.65 |
| Risk factors (Metabolic + Chronic) | | | | | | | |
| | None | 40 (17%) | 4 (11%) | 9 (18%) | 13 (18%) | 14 (17%) | 0.75 |
| | One or more | 202 (83%) | 34 (89%) | 40 (83%) | 60 (82%) | 68 (83%) | |
| Outcome | | | | | | | |
| | Fatality DSO60 | 34 (14%) | 18 (47%) | 6 (12%) | 7 (10%) | 3 (4%) | **<0.0001** |

Values displayed are medians, with IQR in parentheses for continuous variables, or percentages for categorical variables. Percentages were rounded to the nearest unit. Statistical comparison across all four patient clusters, with Kruskal Wallis test for continuous variables, and two-sided χ² test for categorical variables. Source data provided.
Bold values indicate statistical significance *p* < 0.05.

**Table 2 | Comparison of demographics and characteristics of hospital stay between critical clusters 1 and 2 of Discovery cohort with the validation cohort**

| Variables | Entries | Cluster 1+2—Discovery | Validation (entire cohort) | p |
|---|---|---|---|---|
| | N | 87 | 76 | |
| | Age (mean (SD)) | 68.8 (13.72) | 59.42 (13.75) | **<0.001** |
| | Sex (male (%)) | 52 (59.8%) | 60 (78.9) | **0.0084** |
| Max respiratory support throughout hospital stay | | | | |
| | No $O_2$ | 15 (17.2%) | 0 (0%) | **<0.001** |
| | NC/HFNC | 13 (14.9%) | 0 (0%) | **<0.001** |
| | NIV | 16 (18.4%) | 25 (32.9%) | **0.033** |
| | ETI | 42 (48.3%) | 49 (65.3%) | **0.038** |
| | ECMO | 1 (1.2%) | 2 (2.6%) | 0.48 |
| | ICU admission | 56 (64.4%) | 76 (100%) | **<0.001** |
| | Days in ICU (mean (SD)) | 20.5 (20.55) | 14.54 (10.04) | **0.023** |
| Metabolic risk factors | | | | |
| | Any metabolic risk factor | 63 (72.4%) | 50 (75.8%) | 0.36 |
| | Obese (%) | 17 (19.5%) | 27 (40.9%) | **0.022** |
| | Hypertension (%) | 56 (64.3%) | 40 (52.6%) | 0.13 |
| | Diabetes (%) | 34 (39.1%) | 21 (27.6%) | 0.12 |
| Chronic diseases | | | | |
| | One or more (%) | 59 (67.8%) | 38 (50.7%) | 0.12 |
| | Chronic Kidney Disease (%) | 22 (25.3%) | 17 (22.4%) | 0.66 |
| | Heart Failure (%) | 22 (25.3%) | 3 (3.9%) | **<0.001** |
| | Respiratory Disease (%) | 20 (23.0%) | 19 (25.0%) | 0.76 |
| | Liver Disease (%) | 12 (13.8%) | 0 (0%) | **<0.001** |
| | Immunosuppressed (%) | 10 (11.5%) | 10 (13.3%) | 0.75 |
| | Malignancy (%) | 12 (13.8%) | 3 (3.9%) | **0.03** |
| | HIV (%) | 3 (3.4%) | 0 (0%) | 0.10 |
| | Neurological disorder (%) | 11 (12.6%) | 4 (5.3%) | 0.10 |
| Risk factors metabolic or chronic | | | | |
| | One or more (%) | 74 (85.1%) | 58 (89.2%) | 0.16 |
| Outcome | | | | |
| | Fatality DSO60 (%) | 24 (27.8%) | 18 (23.7%) | 0.57 |

Values displayed are medians, with standard deviation in parentheses for continuous variables, or percentages for categorical variables. Percentages were rounded to the nearest unit. Unpaired *t* test for continuous variables, and two-sided $\chi^2$ test for categorical variables. Bold values indicate statistical significance $p < 0.05$.

We next investigated whether there were also differences in plasma viral RNAemia throughout infection (herein referred to as vRNA exposure). 97% of cluster 1's patients had detectable plasma vRNA at least once during their hospital stay, compared to 65% of cluster 2 patients, 55% of cluster 3, and 41% of cluster 4 (Fig. S2F). We fitted a 4-knot spline curve onto the average vRNA per day per cluster and calculated the area under the curve (AUC) per clusters, as a metric for overall exposure (Fig. 2B). Cluster 1's vRNA AUC was significantly greater than that of clusters 2 and 4. Cluster 3 also had significantly greater vRNA exposure compared to cluster 4. Taken together, these results indicate that, among patients with detectable plasma vRNA, those with delayed antibody generation had greater overall exposure to plasma vRNA compared to their severity-matched counterparts.

To better understand how the plasma cytokine and tissue damage profiles evolved, we created a PHATE embedding using the 10 cytokine and tissue damage variables, with days since symptom onset (upweighted, see "Methods" for details) for all available data points within DSO28 of the discovery cohort (*n* = 242 participants, 630 data points, Fig. 2C). Marker color (gradient bar, right) reflects the average cytokines and tissue damage markers concentration of a given sample, unveiling a gradient from the low-concentration region (bottom) to a high concentration one (top). The average trajectories per cluster were plotted atop the embedding (see "Methods" for details). They differed the most in the DSO8-15 interval, with cluster 1 exhibiting the highest cytokine levels and cluster 4 the lowest. We observed convergence of clusters 2,3 and 4 to a common region of the embedding by DSO28, consistent with transition to convalescence. Cluster 1 stood out as maintaining a high concentration of cytokines and tissue damage markers throughout the considered time period, in line with that cluster's greater severity and higher ongoing inflammatory profile compared to the other three clusters.

Therefore longitudinal plasma profiling of hospitalized COVID-19 patients revealed that a delayed generation of RBD-specific antibodies coincided with greater viral exposure throughout the acute phase, suggesting that this delayed antibody response is important to COVID-19 pathogenesis. High and sustained levels of cytokine and tissue damage markers were hallmarks of critical disease.

## Fatal outcome among cluster 1 defined by specific transcriptomic signatures

To understand the molecular features behind the patient endotypes, we analyzed bulk RNA sequencing data from 369 whole blood samples collected within 30 days of symptom onset, 174 of which were collected in the DSO11 timeframe (Supplementary Data 1). Principal component analysis (PCA) on significant differentially expressed genes (DEG—False Discovery Rate (FDR) < 0.01, *n* = 3 271, Supplementary Data 2) between all pairwise comparisons of the four clusters' DSO11-samples revealed segregation of low (1 and 3) and high (2 and 4) antibody clusters along PC1 (Fig. 3A).

PHATE embedding of cluster 1 showed the homogeneous distribution of fatal outcome (Fig. S3A), in line with the absence of differences in plasma levels of vRNA (Fig. S3B), cytokines (Fig. S3C), tissue damage markers (Fig. S3D), and in antibody responses (Fig. S3E) between deceased and survivors of this cluster. PCA on the transcriptomic profiles of the survivor and deceased patients of cluster 1 showed substantial overlap (Fig. 3B), although contrasting both outcomes revealed thousands of DEG (*n* = 1537, FDR < 0.05, |logFC| > 0.5) (Fig. 3C, Supplementary Data 2). Our previously published COVID severity signature[23], aggregated into a single "score" per sample [single sample gene set enrichment analysis (GSEA)—ssCOVID] did not differ between outcomes (Fig. S3F). These results suggest that additional processes, rather than exacerbation of those associated with COVID-19 severity, contributed to fatalities in cluster 1. GSEA using Hallmark gene sets[24] revealed that, within cluster 1, deceased patients had increased signatures of TGFβ and mTOR signaling, while survivors had increased signatures of oxidative phosphorylation, MYC targets, coagulation, DNA repair and fatty acid metabolism (Fig. S3G, Supplementary Data 3). Thus, our results suggest that exacerbated inflammatory responses, when coupled with other mechanisms that include cell metabolism dysregulation, immunosuppression and fibrosis-related signaling (both roles of TGFβ), increase COIVD-19 fatality risk.

## Patients with delayed SARS-CoV-2-specific antibody responses display sustained IFN signaling

To uncover the molecular mechanisms underlying the delayed antibody response, we compared whole blood transcriptomes between clusters according to anti-RBD antibody responses at DSO11. To account for the impact of disease severity on transcriptomic profiles, we performed pairwise comparisons between low and high-antibody patient clusters stratified by clinical status: we compared critical clusters 1 versus 2, and non-critical clusters 3 versus 4. Hundreds of

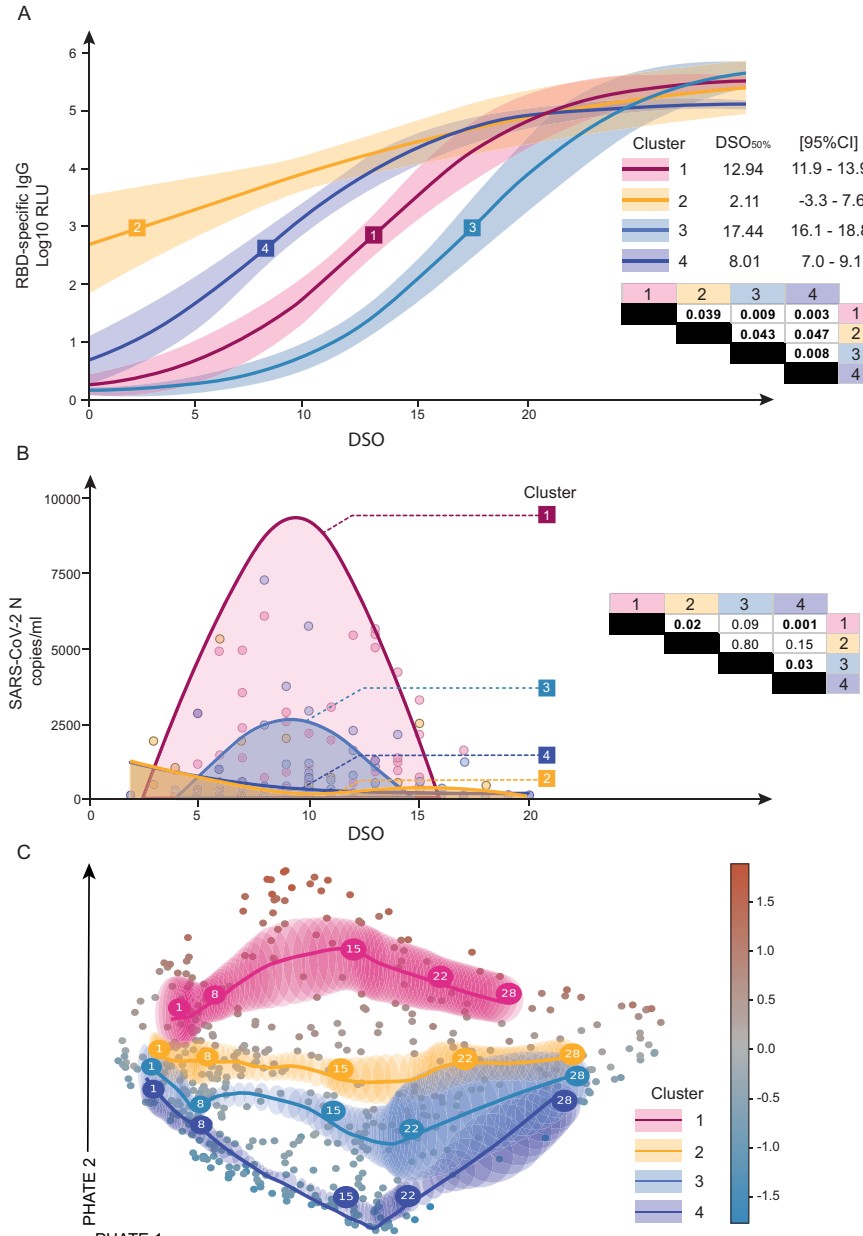

**Fig. 2 | Delayed antibody kinetics is associated with protracted plasma vRNA over a wide range of disease severity. A** Sigmoidal curve fitted to the average per day per patient cluster of RBD-specific IgG responses. The center of the error bars (corresponding to the squares) represent coordinates where 50% of max IgG level is reached per cluster (DSO$_{50\%}$). The 95% confidence intervals (shaded area on graph) and $P$ values (table at bottom right) were calculated using bootstrap comparison of DSO$_{50\%}$. Extrapolated DSO$_{50\%}$ and 95% CI values are on the right of the graph. **B** Model of plasma vRNA detection, fitted to the average per day per patient cluster, among viremic patients only. Bootstrap on the area under the curve (AUC) was used to compare clusters, with $p$ values provided in the table on the right of the graph. Faded dots represent raw data points per DSO. **C** Average trajectory per color-coded patient cluster when the PHATE embedding was performed using cytokines[7], tissue damage markers[3] and DSO across all acute samples. Numbers in large circles represent the day of symptom onset at that coordinate. Shaded area represents confidence interval. Smaller circles in background are datapoints, color-coded by average analytes expression. **A, C** $N = 630$ data points. **B** 224 datapoints (only RNAemia+ participants were considered). **A, B** Two-stage bootstrap, with 1000 simulations. Pairwise comparison between all four clusters. **C** Bootstrapping at the patient level was used to visualize the confidence ellipses representing 3 standard deviations around the average. See material and methods for details. Source data are provided as Source_Data_File.xlsx.

genes were differently associated (FDR < 0.05, |logFC| > 0.5) with antibody status for both comparisons (clusters 1 vs 2 DEGs = 400; clusters 3 vs 4 n DEGs = 674, Fig. 3D, E, Supplementary Data 2). The COVID severity score was increased in cluster 1 compared to all other clusters (Fig. 3F). Various immune signatures were enriched among genes displaying higher expression in the low-antibody response clusters 1 and 3, most notably the IFN gamma response and IFN alpha response pathways (FDR < 2e-4, Fig. 3G, Supplementary Data 3).

Compared to its high-antibody counterpart, cluster 3 had increased signatures of complement and TNFα signaling, and cluster 1 had increased oxidative phosphorylation (Fig. 3G). Gene ontology enrichments on the Biological processes[25] similarly showed that patient clusters with delayed antibody responses were enriched for pathways related to IFN signaling and to the defense against invading pathogens (Fig. S3H, Supplementary Data 3), with cluster 3 further enriched for pathways related to regulation of cytokine production and

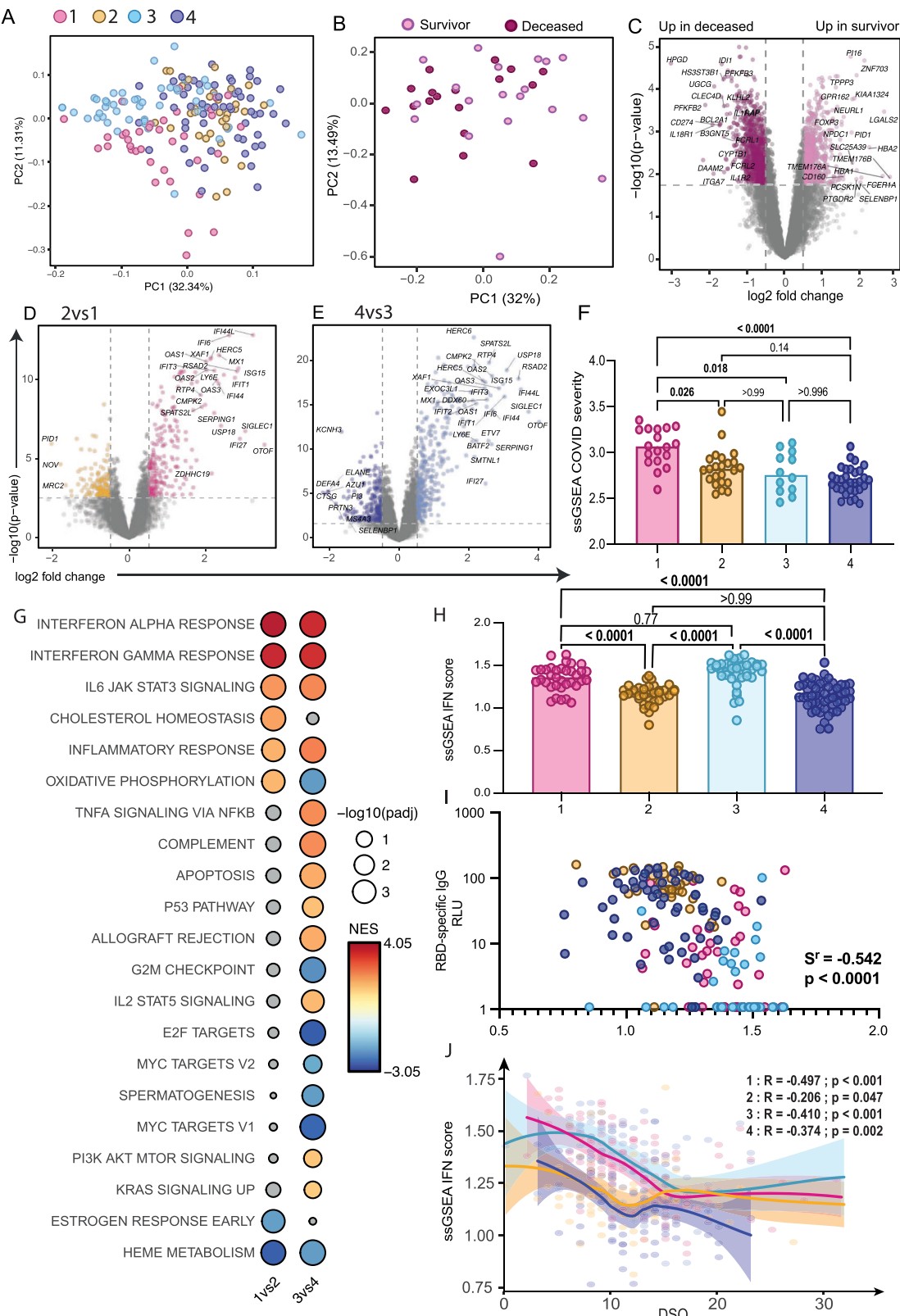

TLR7 signaling (Fig. S3I, Supplementary Data 3). Overall, the transcriptomic profile of low-antibody clusters was characterized by heightened defense pathways against pathogens and type I and II IFN signatures.

To see whether the association between these pathways were reproducibly associated to delayed antibody production, we performed bulk whole blood RNA Sequencing on a subset of samples prospectively collected within the validation cohort (V1 $n = 18$; V2 $n = 14$). We identified 48 DEG (FDR < 0.05, |logFC| > 0.5) between both clusters, the lower number being in line with the smaller size of the clusters. The high fatality cluster was again enriched in the interferon response pathways (Fig. S3J). It was also enriched in other pathways associated to the discovery clusters with delayed antibody responses: TNFa signaling via NFkB, inflammatory response and IL6 JAK

**Fig. 3 | Patients with delayed SARS-CoV-2-specific antibody responses display sustained IFN signaling. A** Principal component analysis (PCA) based on significant DEG (FDR < 0.01, n = 1346 genes) from all pairwise comparisons across the 4 patient clusters. Each dot represents a separate patient, sampled at DSO11, and color-coded to their respective cluster. **B** PCA on whole transcriptome (n = 10,236 genes) of patients in cluster 1 only at DSO11, color coded by survival or fatal outcome at DSO60. **C** Volcano plot of differentially expressed genes (DEG) based on outcome, with significant genes color-coded (FDR < 0.05; |logFC| > 0.5). Dashed lines represent the nominal p-values corresponding to an FDR = 0.05, and points with an FDR < 0.05 are highlighted in color. Mauve dots represent genes increased in fatal outcome, and pink, genes increased in survivors. Relevant genes are tagged. **D, E** Volcano plots of contrasts (**D**) 1 vs 2 or (**E**) 3 vs 4, with significant genes (FDR < 0.05; |logFC| > 0.5) color-coded and relevant genes tagged. **F** Single sample (ss)GSEA of published COVID-19 severity score[23] across patient clusters. **G** GSEA using Hallmark dataset on t-statistics from aforementioned contrasts. Red dots are pathways enriched in the low antibody clusters 1 and 3 compared to 2 and 4, respectively, while blue dots are pathways enriched in high antibody clusters. Significant hits are colored. Size of the circle is representative of significance of enrichment. **H** ssGSEA IFN score calculated from the combination of the "interferon gamma response" and "interferon alpha response" Hallmark gene sets across patient clusters. **I** Correlation between IFN score and contemporaneous RBD-specific IgG levels at DSO11. **J** ssGSEA IFN score over time (DSO < 40) per patient cluster, with confidence intervals shaded. R and p values of each cluster are annotated at the bottom of the figure. *N*: 1 = 100; 2 = 93; 3 = 86, 4 = 98 (377 in total). **C, D, E** P-values were obtained from least squares linear regression models (two-tailed). False discovery rates were calculated using a permutation-based approach that derives the null empirically. **F, H** Kruskal-Wallis with Dunn's multiple comparison tests. Adjusted p-values are shown. **G** fgsea p-values were calculated using a permutation-based approach. Multiple testing correction was performed using the Benjamini-Hochberg method. **I, J** Two-tailed Spearman correlations. Medians are shown in bar charts. Source data are provided as Source_Data_File.xlsx (ssGSEA scores), as well as in Supplementary Data 1, 2 and 3.

STAT3 signaling. These results further support and validate the association between the IFN signatures and the delayed antibody response.

Within these interferon signatures, we found that certain interferon-related genes are more associated with severity than others. Genes known to be involved in MHC class II antigen presentation and processing, such as CD74, CD86, HLA-DRB1, and genes coding for interleukin receptors that serve to inhibit proinflammatory cytokines, such as IL10RA and IL18BP, are more highly expressed in less severe patients (Fig. S3K). In contrast, other genes, including CD274 (PD-L1), FCGR1A, SOCS3, TNFAIP6, and UPP1, display higher expression in critical patients compared to less severe patients (Fig. S3L). Most of these latter genes are involved in immune- or inflammation-related processes, highlighting that, within interferon signaling, it is those associated with inflammation that are related to disease severity.

To further investigate these IFN signatures, we next performed single sample (ss)GSEA combining all genes of both IFN gamma/alpha response pathways into a single score (ssIFN, Fig. 3H). In line with the enrichment analyses, clusters 1 and 3 had significantly higher ssIFN scores than clusters 2 and 4, respectively. The ssIFN score showed a strong negative correlation with contemporaneous RBD-specific IgG levels (Fig. 3I), and showed weaker positive correlations with contemporaneous plasma vRNA (Fig. S3M), cytokine scores (Fig. S3N), and tissue damage scores (Fig. S3O). Thus, sustained IFN responses are associated with multiple immunopathological traits in COVID-19.

We next plotted the IFN score over days since symptom onset per patient cluster (Fig. 3J). This analysis revealed that clusters 1 and 3 had high and sustained IFN for a longer period than the two other patient clusters, although all clusters converged around DSO15.

These results show that sustained upregulation of IFN pathways was associated with the delayed generation of SARS-CoV-2-specific antibody responses, this in patients exhibiting a wide range of disease severity.

### High IFN signaling is negatively associated with blood RBD-specific B cell and plasmablast frequencies

To investigate the cellular basis of poor antibody responses in the low-antibody patient clusters, we examined the blood SARS-CoV-2-specific B cell and plasmablast (PB) populations (Fig. S4A). Staining peripheral blood mononuclear cells (PBMCs) with two fluorescently-labeled recombinant RBD probes identified RBD-specific B cells (Fig. 4A) and PB (Fig. S4B). While RBD-specific B cells were detectable in convalescent patients (Fig. S4CD), RBD-specific PB were only detectable during acute infection (Fig. S4EF), in line with the kinetics of circulating PB in COVID-19[26]. To account for lymphopenia in patients with COVID-19[26], we used contemporaneous clinical complete blood counts (CBC) to calculate the absolute frequencies of RBD-specific B cells and PB per mL of blood. Both populations correlated positively with each other (Fig. 4B) and with RBD-specific IgG levels (Fig. 4C, D), consistent with their role in antibody production. B cell frequencies negatively correlated with the contemporaneous ssIFN score (Fig. 4E), but no significant correlation was found for the PB (Fig. 4F). Neither population correlated with the ssCOVID severity score (Fig. S4GH). The cluster-level patterns were consistent with the strength of the IFN signatures (Fig. 4G): cluster 3 had lower counts, and clusters 2 and 4 greater counts of SARS-CoV-2-specific B cells. For PB, differences did not reach statistical significance, as the spread within clusters was more pronounced (Fig. 4H). There were no differences between acute infection clusters in isotype expression by RBD-specific B cells, which were mostly IgM+ and/or IgG+ (Fig. S4I). This pattern differed from a separate cohort of convalescent outpatients (DSO > 100), in whom RBD-specific B cells were almost exclusively IgG+. RBD-specific PB displayed similar trends, albeit with a greater representation of IgG/IgA double-positive than their B cell counterparts (Fig. S4J).

Taken together, these results suggest that the sustained IFN signaling may hamper the generation of SARS-CoV-2-specific B cells, and consequently delay antibody responses in patients with various clinical presentations.

### High IFN signaling is negatively associated with Spike-specific CD4+ T cell responses

As persistent IFN signaling impairs adaptive virus-specific T helper immunity in murine models[27,28], we next examined the links between IFN transcriptional signatures and development of SARS-CoV-2-specific T cell responses. During acute SARS-CoV-2 infection, immunodominant peptides recognized by CD4+ T cells were mainly in the Spike-derived peptide pool, which also elicited CD8+ T cell responses in a majority of patients (Fig S5A–C). We measured the T cell responses against Spike using an activation-induced marker (AIM) assay we previously described[29]. Spike-specific CD4+ T cells were detected by co-upregulation of CD69 with CD40L or OX40. We used a Boolean OR gating strategy to include overlapping populations (Figs. 5A, S5D). We again calculated absolute counts of Spike-specific CD4+ T cells by using CBC.

Spike-specific CD4+ T cells were detectable in most acute and all convalescent samples (Fig. S5E). These frequencies correlated negatively with the ssIFN score (Fig. 5B), suggesting a negative impact of IFN signaling on T helper responses. There was no correlation between CD4+ T cell responses and ssCOVID severity scores (Fig. 5C). Spike-specific CD4+ T cells counts positively correlated with RBD-specific B cells counts (Fig. 5D), in line with the role of CD4+ T cell help in B cell immunity. No significant correlation was observed between T helper responses and RBD-specific PB (Fig. 5E). Spike-specific CD4+ T cell responses were detected in most patients in all clusters, except for cluster 3 (Fig. 5F), underscoring a defect in adaptive T helper responses and paralleling the defects identified for RBD-specific B cells.

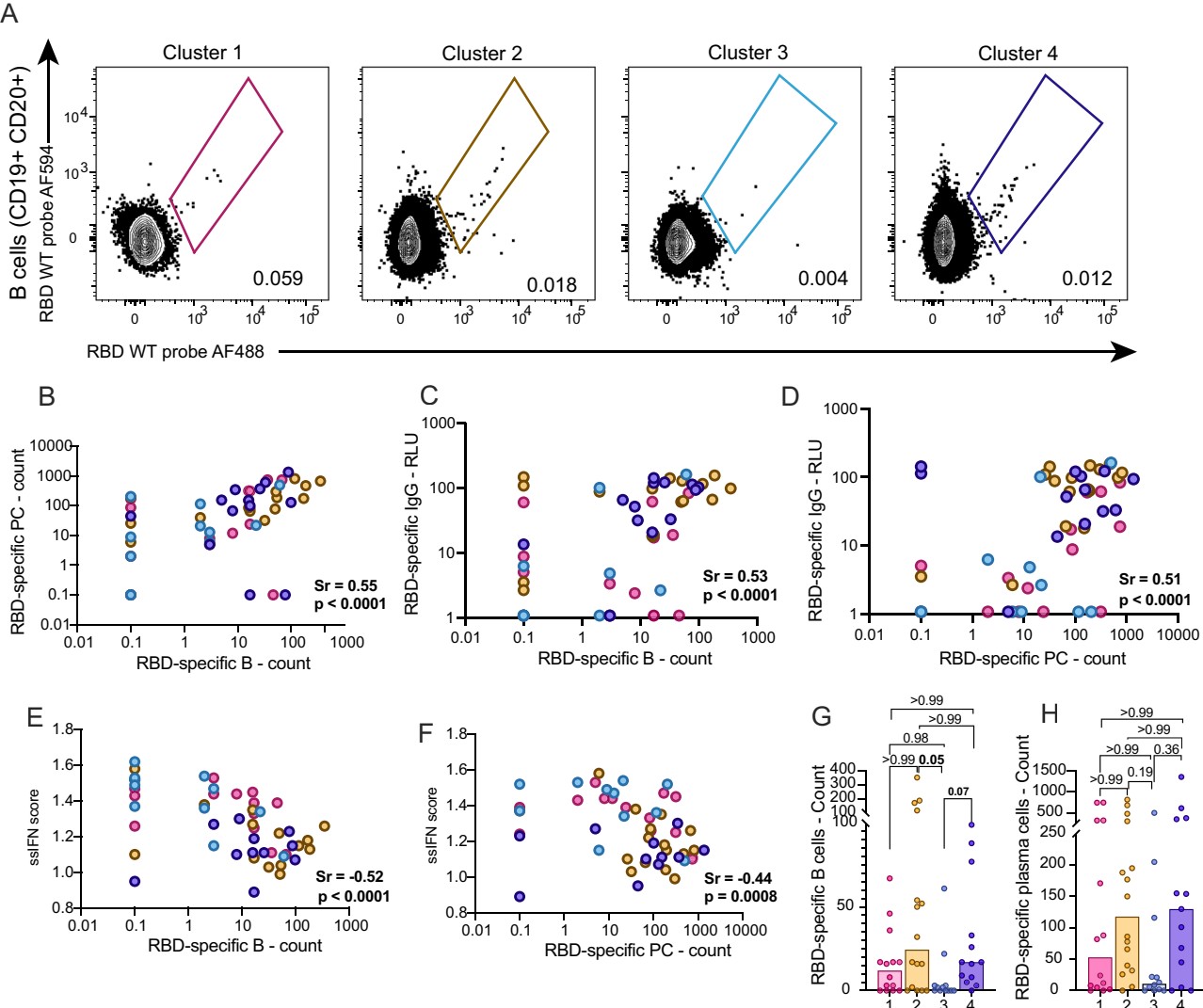

**Fig. 4 | Elevated IFN signaling is negatively associated with RBD-specific B cell and plasmablast frequencies. A** Representative flow cytometry plots of RBD-specific B cells identified per patient cluster at DSO11. **B**–**F** Correlation between (**B**) absolute counts of RBD-specific B cells and RBD-specific PB; (**C**) absolute counts of RBD-specific B cells and RBD-specific plasma IgG levels; (**D**) absolute counts of RBD-specific PB cells and RBD-specific plasma IgG levels; (**E**) absolute counts of RBD-specific B cells and ssGSEA IFN score; (**B**) absolute counts of RBD-specific PB cells and ssGSEA IFN score. **G**, **H** Per patient cluster, absolute counts of RBD-specific (**G**) B cells or (**H**) PB. n for cluster 1 = 14; 2 = 16; 3 = 12; 4 = 13. **B**, **C**, **D**–**F** Two-tailed Spearman correlations. **G**, **H** Kruskal-Wallis with Dunn's multiple comparison tests. Adjusted $p$-values are shown. For patients with undetectable RBD-specific B and/or PB counts, they were assigned value 0.1. Medians are shown in bar charts. Source data are provided as Source_Data_File.xlsx.

We measured Spike-specific CD8[+] T cells by their co-upregulation of CD69 and 41BB (Fig. 5G). Spike-specific CD8[+] T cells were also detectable in most acute and all convalescent samples, although they were less frequent than their CD4[+] T counterparts (Fig. S5F). They correlated neither with ssIFN nor with ssCOVID scores (Fig. 5HI), nor did they differ across clusters (Fig. 5J). Despite the differential association of Spike-specific CD4[+] and CD8[+] T cell responses with ssIFN signatures, the two cell populations themselves correlated positively (Fig. 5K), consistent with the role of CD4[+] T cell help in primary CD8[+] T cell responses.

Taken together, these results suggest that sustained IFN signaling negatively impacts SARS-CoV-2-specific CD4[+] T cell responses, which in turn hamper the generation of SARS-CoV-2-specific B cells.

## Discussion

Through a relatively simple immunovirological plasma profile 11 days after symptom onset (14 analytes: RNAemia, seven cytokines, three issue damage markers and three RBD-specific antibody

isotypes), we identified COVID-19 patient endotypes with important differences not only in disease severity and outcome, but also in the quantity and timing of innate, antibody, and cellular responses to SARS-CoV-2. We also found that excessive IFN signaling likely contributes to differential kinetics of virus-specific antibody, B cell and CD4[+] T cell responses. Early robust antibody and T cell immunity, coupled with a low inflammatory profile and low plasma viremia, is associated with moderate disease and good prognosis (cluster 4). When robust SARS-CoV-2-specific immune responses are maintained in the setting of a higher inflammatory profile, as typically observed in critical disease, the prognosis is still good (cluster 2). Patients with sustained IFN signaling also have delayed antibody and CD4[+] T cell responses, which can be associated with a good prognosis if viremia and inflammation are low or moderate (cluster 3). However, sustained IFN signaling coupled with high viremia, exacerbated inflammatory profile and low SARS-COV-2-specific B cell and CD4[+] T cell responses is associated with the distinctively high-fatality cluster 1.

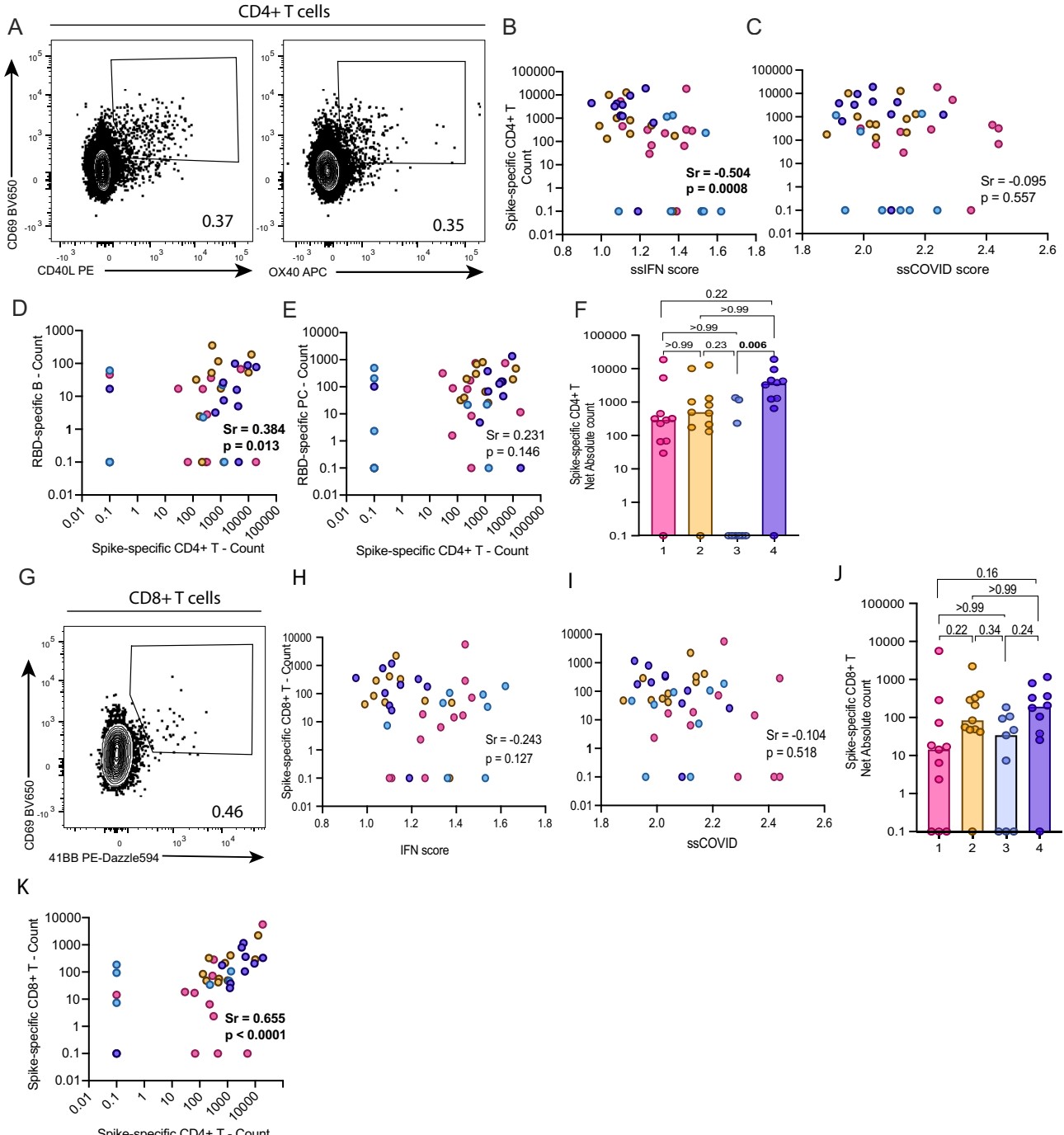

**Fig. 5 | Elevated IFN signaling is negatively associated with Spike-specific CD4+ T cell responses. A** Representative flow cytometry gates used to detect Spike-specific CD4+ T cells following 15 h peptide stimulation. Boolean OR gating strategy used. **B–E** Correlations between absolute counts of Spike-specific CD4+ T cells with (**B**) ssGSEA IFN score; (**C**) ssGSEA COVID-19 severity score; (**D**) absolute counts of RBD-specific B cells or (**E**) absolute counts of RBD-specific plasma cells. **F** Comparison of absolute counts of Spike-specific CD4+ T cells per patient cluster. **G** Representative flow cytometry gate used to detect Spike-specific CD8+ T cells

following 15 h peptide stimulation. **H, I** Correlations between absolute counts of Spike-specific CD8+ T cells with (**H**) ssGSEA IFN score and (**I**) ssGSEA COVID-19 severity score. **J** Comparison of absolute counts of Spike-specific CD8+ T cells per patient cluster. **K** Correlation between the absolute counts of both Spike-specific T cell populations. n for cluster 1 = 11; 2 = 11; 3 = 9; 4 = 10. **B–E, H, I, K** Two-tailed Spearman correlation. **F, J** Kruskal-Wallis with Dunn's multiple comparison tests. Adjusted *p*-values are shown. Medians are shown in bar charts. Source data are provided as Source_Data_File.xlsx.

Among critically ill patients, our approach delineated the subset of individuals at very high risk of fatality from those with unexpectedly good prognosis with greater accuracy compared to binning patients based on clinical severity, underlining the advantage of using immunovirological endotypes. The cluster-based method combined with model fitting and bootstrap pairwise comparisons

made up for the sparsity of data points per patient, allowing for robust comparisons of trajectories in a time-dependent manner. The findings were highly reproducible in a separate cohort located in a different country, despite differences in the laboratory methods used. This robustness of the k-means algorithm, which builds on the high-dimensional relationships of features rather than absolute

values, can be a major asset in multicentric translational biomedical research, where ensuring reproducibility of data can be complicated by the use of diverse technical platforms (e.g., institutional clinical lab instruments). For instance, despite measuring antibodies targeting different SARS-CoV-2 proteins (RBD for discovery; N for validation), both cohorts displayed low responses at DSO11 among the high fatality clusters. These results suggest an effect on the antibody response against SARS-CoV-2 as a whole, and underscore the value of a multiparametric approach in deciphering patient heterogeneity.

The RBD-specific antibody trajectory analyses revealed that the low antibody levels observed at DSO11 in clusters 1 and 3 were due to a delayed initiation of the antibody response to RBD, rather than an inability to do so. Indeed, the modeled curves converged prior to DSO30, consistent the robust SARS-CoV-2 specific antibody responses in convalescent individuals after critical disease. One unavoidable limitation here is possible survivorship bias, as we cannot determine if the critical cases who succumbed early in disease course had the potential to mount this response. These granular analyses also differentiate vRNA kinetics that do not merely align with clinical severity and were not identified by more traditional statistical tools[1,30]. While vRNA loads were highest in the high-fatality cluster 1, they were also sustained in the low-antibody, non-critical cluster 3, whereas both clusters that rapidly developed RBD-specific antibodies (2 and 4) readily achieved viral clearance. These results are consistent with the critical role of antibody responses in controlling viral replication. In outpatients, anti-Spike neutralizing monoclonal antibodies and antiviral drugs decrease the risk of disease progression only when given early after symptom onset[31,32]. There is also no or limited impact of monoclonal antibodies in people hospitalized for COVID-19[33], except for a subset of those who have not yet seroconverted[34]. These data indicate that once underway, the pathogenic inflammatory cascades, which diverge as early as DSO8, have limited sensitivity to these interventions.

Whole-blood transcriptional profiling provided important insight into the molecular features underpinning these endotypes. While the hierarchy of the COVID-19 severity signature we previously established[23] followed the clinical profile, pair-wise comparisons between the two critical clusters (1 vs 2) and between the non-critical clusters (3 vs 4) revealed differences in gene expression beyond disease severity. The pathways upregulated in low-antibody clusters 1 and 3, including IL-6-JAK-STAT3 signaling and other inflammatory pathways, are those for which blunting through therapies results in survival benefit (specifically by tocilizumab, sarilumab and baricitinib; and broadly by dexamethasone)[4,9,11], supporting a mechanistic explanation for these interventions. Signatures of complement activation and signaling through TLR7 and TNFα were only upregulated in cluster 3, suggesting that innate immunity may contribute to a moderate disease course despite sustained IFN signaling. Conversely, fatal cases of cluster 1 had further upregulation of IL-6-JAK-STAT3 signaling and inflammation, with exacerbated TNF signaling compared to cluster 1's survivors. They also exhibited lower expression of MYC targets, which include many proliferation and anti-apoptotic pathways, along with depressed metabolic pathways. These findings suggest a disruption of key cellular processes in patients who subsequently succumb to their illness, which may help guide investigations of new therapeutic targets.

Because of potential mechanistic implications, a key finding of the transcriptional analyses is the differential kinetics of both type I and II IFN signatures among patient clusters. Although pronounced in cluster 1 fatalities, exacerbated IFN signatures did not merely coincide with disease severity, but rather with impaired generation of SARS-CoV-2-specific antibody, B cell and CD4+ T cell responses. These defects are probably causally linked, given the critical role of CD4+ T cell help for B cells[35]. In contrast, we observed no significant associations with CD8+ T cell responses. These patterns suggest that protracted activation of

IFN pathways can adversely affect some anti-SARS-CoV-2-specific responses, in addition to impairing repair mechanisms of damaged lung tissues[36,37]. One potential mechanism would be the decreased efficacy of antigen presentation in the presence of high interferon signaling[38], in line with the negative association those genes have with severity in our data. Seminal studies in the murine lymphocytic choriomeningitis virus (LCMV) model support this hypothesis[28,39]: while IFNs are critical in the early generation of antiviral responses, sustained type I IFN signaling in chronic Clone 13 infection was associated with poor antibody, B cell and CD4+ T cell responses. In this context, blockade of type I interferon signaling by an anti-IFNAR1 antibody decreased viral loads and improved immune responses. A similar benefit has recently been observed though the modulation of type I IFN in SARS-CoV-2 infection of rhesus macaques[40]. This dual role of IFN, where timing, rather than quantity, is central to an appropriate adaptive response, is dubbed "The Interferon paradox"[41], and is also supported by data in SIV infection of non-human primates[42]. In conjunction with data supporting a protective role of IFNs early in COVID-19 course[12,13], our results suggest this paradox is operative in SARS-COV-2 infection in defined patient subgroups (even though, in contrast to LCMV Clone 13 and SIV, this major human viral disease is—with rare exceptions—an acute viral infection). This explains discrepancies around IFN in the literature. Individuals with inborn errors in type I IFN responses[13], genetic variants which lower IFN responsiveness[43] or preexisting anti-IFN autoantibodies[12] are at greater risk of severe COVID-19 because they never have the initial IFN (key for reducing viral replication and the generation of an initial anti-viral response). However, the IFN signature should drop quickly, or it hampers the antiviral response and causes immunopathology. Patients are often put on mechanical ventilation around 9 days after symptom onset, so studies comparing critical and non-critical cases enrich in patients at those times, which is why high IFN signal was associated to severe disease in blood transcriptomics[44] and lung in situ[45] studies. It also explains the results from the clinical trial with IFN[14]: it was administered too late to have any beneficial effect, and only the few people with genetic defects in the IFN pathway would have benefited from it.

In summary, we show that SARS-CoV-2-infected patients experiencing high, sustained IFN signaling have a delayed generation of Spike-specific CD4+ T cells and RBD-specific B cells. This directly links to a delay in the antibody response against the virus and, in patients also presenting increased inflammation, tissue damage, and plasma RNAemia, is associated with a highly fatal profile. Compared to mechanistic studies in mice, a weakness of the present observational study is the lack of direct manipulation of the type I IFN pathway. However, our results can have direct clinical relevance, and at least in part explain why clinical trials of recombinant IFN therapy have yielded disappointing results in COVID-19[46]. While our study lacked investigation of the lung compartment, others also support a pathophysiologic role of excessive type I IFN in this organ[16,36]. Hence, excessive type I IFN signaling is likely detrimental at multiple levels that involves adaptive immunity, and tissue repair. Whether targeted blockade of IFN pathways—rather than IFN supplementation—might be beneficial in specific subgroups of patients with COVID-19 would require further investigation.

## Methods

### Participants and samples

We investigated prospectively COVID-19 individuals hospitalized between April 2020 and August 2021 with symptomatic infection with a positive SARS-CoV-2 nasopharyngeal swab (NSW) reverse-transcription polymerase chain reaction (RT-PCR) who were admitted to the Centre Hospitalier de l'Université de Montréal (CHUM) or the Jewish General Hospital (JGH) and recruited into the Biobanque Québécoise de la COVID-19 (BQC19)[47] Blood draws were performed at baseline and, when consistent with patient care, at 2, 7, 14 and 30 days

(±3 days) after enrollment. Exclusion criteria were breakthrough or reinfection, plasma transfer therapy (could change plasmatic profile), or vaccination prior to infection. The study was approved by the Research Ethics Board of the Jewish General Hospital (JGH) and the Comité d'Éthique à la Recherche (Research Ethics Board) du Centre Hospitalier de l'Université de Montréal (CHUM) (multicentric protocol for the BQC19 biobank: MP-02-2020-8929; local protocol for the specific study: 20.169) and written informed consent obtained from all participants or, when incapacitated, their legal guardian before enrollment and sample collection. Research adhered to the standards indicated by the Declaration of Helsinki. Blood draws were also performed on 50 asymptomatic, SARS-CoV-2 antibody negative uninfected controls (UC), early in the pandemic (spring 2020). COVID-19+ hospitalized patients were stratified based on the severity of respiratory support at the DSO11 timepoint: critical patients required mechanical ventilation [noninvasive ventilation, endotracheal intubation, extracorporeal membrane oxygenation – (ECMO)], and non-critical patients, encompassing patients with moderate disease required no supplemental oxygen and patients with severe disease requiring oxygen supplementation by nasal cannula. Mortality was followed up to DSO60. Medical charts were reviewed by physicians and study coordinators for data collection on demographics, co-morbidities, risk factors, severity state, time of infection, etc. (see Table 1). Median age of the UC cohort was 37 years (range: 24–57), and 30 individuals were males (60%). Clinical data were collected within hospital units by the clinical teams and clinical research teams, as part of standard patient care (and therefore included in the electronic medical record) or to fill in some specific fields of the case report form. Patient and sample identifiers were created in our research group and are not known to anyone outside our research group, as to protect the identity of the study participants. All samples were biobanked and conserved at −80 °C (for plasma) or in the gas phase of liquid nitrogen (for PBMCs).

The validation cohort was recruited at Uppsala University Hospital in Sweden. The study was approved by the Swedish National Ethical Review Agency (Pronmed study; 2017-043, amended 2019-00169, 2020-01623, 2020-05730 and 2022-00526-01) and registered a priori at ClinicalTrials.gov (NCT03720860). Informed consent was obtained from the patient or next of kin if the patient was unable to give consent. The Declaration of Helsinki and subsequent revisions were followed. The study included 123 adult patients admitted to intensive care during the first wave of the pandemic between March 15th, 2020, and July 14th, 2020. All patients had confirmed SARS-CoV-2 by RT-PCR from NSW. Exclusion criteria were pregnancy, currently breastfeeding, and age under 18. A validation cohort of 76 patients was collected from the Pronmed study biobank with analyses that matched the parameters used for PHATE embedding in the discovery cohort. All samples were biobanked and conserved at −80 °C (for plasma) or in the gas phase of liquid nitrogen (for PBMCs).

**Ethics and inclusion statement**. Local researchers from Montreal were included throughout the research process. As it focused on identification of high-risk cases following COVID-19, it is locally-relevant research. The expected contributions of each collaborator was decided upon during study design. This study was done in collaboration with the BQC19 biobank (https://www.bqc19.ca/en), and our citations reflect that.

With the inclusion of a validation cohort, researchers from Upsalla, Sweden, oversaw recruitment of local collaborators as they saw fit. Given the urgency of COVID-19 research during the pandemic, agreements between institutions were expedited, which allowed us to partially share data. Although there was no evidence that SARS-CoV-2 could be transmitted from blood, all research performed on blood was performed in BSL2*, with the use of N-95 and protective splash guards.

The study was approved by local ethics committee, as stated previously. For both cohorts, patient and sample identifiers were created in our research group and are not known to anyone outside our research group, as to protect the identity of the study participants.

## Measurements of plasma analytes
**Quantification of plasma SARS-CoV2 RNA**. For the discovery cohort, absolute copy numbers of SARS-CoV-2 RNA (N region) in plasma samples were measured by real-time PCR. Total RNA was extracted from 230 µL of plasma collected on acid citrate dextrose (ACD) tubes using the QIAamp Viral RNA Mini Kit (Qiagen Cat. No. 52906). Two master reaction mixes with specific primers and probes were prepared for quantification of N gene from SARS-CoV-2 and 18 S (as a control for efficient extraction and amplification). N SARS-Cov2 quantifications were performed in quadruplicate and 18 S measurements were performed in duplicate. A positive and no-template negative controls were included in all experiments. Purified RNA N transcripts (1328 bp) were quantified by Nanodrop, and the RNA copy numbers were calculated using the ENDMEMO online tool (see "STAR methods" for details).

For the external validation cohort, plasma viral RNA was determined by real-time RT-PCR recognizing the SARS-CoV-2 N-gene using the 2019-nCoV N1 reagent based on the Center for Disease Control (CDC) of the United States protocol as described previously[3].

**Measurements of cytokines, chemokines and tissue damage markers**. The analytes measured are listed in Supplementary Table S1. For the discovery cohort, duplicates of SARS-CoV-2-inactivated plasma samples were analyzed using a customized Human Luminex Discovery Assay (LXSAHM-26, R&D Systems). Datasets were acquired on two separate machines (BioPlex, MagPix), with 30 repeat samples performed on both. Linear regression was performed for each analyte and regressions used for batch correction of samples acquired on the BioPlex. As PHATE requires complete datasets, some analytes with low sensitivity that could not be corrected were excluded: CCL20, CCL3, CCL7, IFNα, GM-CSF, IL-10, IL-17A, IL-1b, IL-2, and IL-33. We retained all analytes significantly associated with fatal outcome ($p < 0.01$) in our previous work[1]: TNFα, CXCL13, IL-6, IL-23, CXCL8/IL-8, angiopoietin-2, RAGE, and Surfactant Protein D. For the validation cohort, plasma cytokines were measured using citrated plasma samples for 27 biomarkers with the Bio-plex assay using a Luminex MagPix instrument (Bio-Rad Laboratories AB, Sundbyberg, Sweden) as described previously[6]. Of these 27, only the analytes in common with those measured in the discovery cohort were retained for the clustering and the PHATE embedding: TNFα, IL-6, CXCL8, IL1Ra, and CCL2. Plasma RAGE was also measured (and included as input to k-means and PHATE) using the Proximity extension assay (PEA) at the Clinical Biomarkers Facility (SciLifeLab, Uppsala, Sweden) using the Cardiovascular panel from OLINK Proteomics® (Uppsala, Sweden).

**Antibody measurements**. In the discovery cohort, RBD-specific IgG, IgM and IgA were quantified using an in-house SARS-CoV-2 RBD ELISA assay, as described elsewhere[48]. Plasma from pre-pandemic uninfected donors were used as negative controls to calculate the seropositivity threshold in the ELISA assay. The monoclonal antibody CR3022 (RRID: AB_2848080, from Dr M. Gordon Joyce)[49] was used as a positive control. The seropositivity threshold was established using the following formula: mean of all COVID-19 negative plasma + (3 standard deviation of the mean of all COVID-19 negative plasma).

In the validation cohort, nucleocapsid-specific antibody levels at DSO11 were measured using the NovaLisa® SARS-CoV-2 IgA, IgM, and IgG kits according to manufacturer's instructions (COVA0940, COVM0940, COVG0940, Novatec Immundiagnostica, Dietzenbach, Germany), and at DSO 20 using by FluoroEnzymeImmunoassay (FEIA), Phadia AB, Uppsala, Sweden as described previously[17].

## Data dimensionality reduction and clustering

**PHATE**. Dimensionality reduction is a necessary step to visualize and explore high dimensional datasets. While PCA[50] is commonly used, the resulting components are restricted to a linear projection of the input data, thereby limiting the expressiveness of the resulting visualizations. Recent advances in dimensionality reduction techniques instead favor so-called manifold learning algorithms, such as PHATE (Potential of Heat-diffusion for Affinity-based Trajectory Embedding), which can compute a nonlinear transformation of the data to effectively represent the latent structure of a dataset in low dimensions[22]. PHATE begins by computing a sample-sample affinity graph, i.e., a graph connecting pairs of similar samples to form "neighborhoods". The graph can then be leveraged to compute transition probabilities−the probability of a sample "jumping" to one of its' graph neighbors in a random walk. By iteratively repeating this operation in a process known as diffusion, transition probabilities can be derived for any pair of samples in the dataset and therefore represent a useful notion of pairwise similarity, with probability transitions being high between close samples in terms of the diffusion geometry. Armed with these similarities, PHATE computes a two-dimensional embedding where the Euclidean distance reflects the dataset's intrinsic structure as captured by diffusion, thus enabling the interpretation and analysis of high dimensional data Data samples are subsequently embedded in a low dimensional space (usually, 2 dimensions for visualization on scatter plots) by preserving both the local and long-range pairwise similarities, meaning the distance between "neighborhoods" in the embedding are meaningful. Intuitively, this can be thought as « unfolding » the sample-sample graph in low dimensions while preserving the graph's intrinsic structure, as captured by diffusion affinities.

In practice, only explanatory variables (in our case, plasma concentrations) are used as input to PHATE and the structure of the embedding is therefore entirely unsupervised. PHATE then generates informative low-dimensional representations of the data and is known to preserve substructures of interest−such as clusters−while being robust to noise and non-uniform sampling of the underlying data manifold. Any variable of interest−such as patient outcome or clinical data−can be used for coloring. Particularly, explanatory variables can be used as color gradients to observe how they are distributed on the visualization (e.g., to identify high antibody and low antibody neighborhoods).

**DSO11 PHATE embeddings.** We computed PHATE visualizations of cross-sectional samples taken 11 days after symptom onset (DSO11 + / − 4 days) in the discovery cohort ($n = 242$, 14 measurements) and the validation cohort ($n = 76$, 10 measurements). Samples with missing measurements were removed, and if a given patient had more than one sample in the considered time period, only the one closest to DSO11 was included. Each 2D marker in the resulting scatter plots therefore summarizes the plasma profile of a single patient. We use standard scaled (mean 0, variance 1) log concentrations of each sample as input to the PHATE Python package (v1.0.9) with a *knn* parameter of 10 for the discovery cohort and of 5 for the validation one. Both cohorts use a diffusion time *t* parameter of 50.

**DSO11 K-means clustering.** K-means[51] clustering aims to partition samples into different clusters with high intra-cluster similarity. The identified clusters can then serve as a basis for comparing typical groups or sample profiles. The method represents clusters as centroids (cluster centers), and iteratively refines them by alternating two steps: (1) assign samples to the closest centroids, and (2) replace centroids with the per-cluster sample means based on current assignments. To mitigate limitations of K mean relating to local minima and centroid initialization, our code runs 10 initializations and picks the best one.

We clustered the same samples as the ones used for the DSO11 PHATE embeddings, using the k-means implementation of the scikit-learn[52] Python package (v1.0.2), using Euclidian distances. We used 4 clusters for the discovery cohort and 2 clusters for the validation cohort due to the smaller number of samples. Data preprocessing was identical to the process used for PHATE.

**MELD.** Of particular interest to visualize binary outcome variables is the MELD algorithm[53], which performs a low-pass filtering of binary variables over the sample-sample graph to make them "smoother" over neighborhoods. The resulting values are used to compute relative likelihoods, thereby indicating if some groups of similar samples are enriched or depleted in a specific condition. In practice, this can be used to turn a binary variable into a continuous gradient which can be visualized on top of a PHATE embedding. We used MELD (v1.0.0) to obtain smooth visualizations of critical severity and fatal outcome in the DSO11 discover cohort.

**Longitudinal embedding.** We visualized the evolution of cytokine and tissue damage profiles in the discovery cohort using a second PHATE longitudinal embedding. The time horizon was increased to include all samples from DSO0 to DSO28. Only samples from patients selected for the DSO11 analysis were considered to better understand the progression of the identified DSO11 subgroups in the discovery cohort. Contrary to the previous embedding, each resulting 2D marker reflects a sample ($n = 491$) and the same patient can be represented multiple times. To emphasize temporal structure, DSO was used as an input for PHATE, in addition to the 10 cytokine and tissue damage log concentrations. We again centered the data (mean 0) and apply standard scaling (variance 1). We then upweighted the time variable by a factor of $\sqrt{8}$ in distance computations in PHATE to better visualize time, as suggested in[22]. The resulting PHATE embedding is colored by the average of the 10 standard scaled log concentrations used as input.

To visualize the evolution of the DSO11 clusters, we computed one multivariate linear regression on the samples of each cluster using DSO as the explanatory variable and the 10 log concentrations as a multivariate response. The linear model of each cluster was used to obtain continuous log concentration predictions for the DSO0-28 range. The resulting log concentration curves were projected onto the 2D precomputed longitudinal embedding using interpolation with existing samples, as implemented in the PHATE Python package. Two-stage bootstrap (see "Statistical analyses") was used to visualize the confidence ellipses representing 3 standard deviations around the average.

## Statistical analyses

**Statistical comparisons of single variables.** The type of statistical test is specified in the figure legends. Given the size of the cohorts, we opted for conservative non-parametric tests. Mann−Whitney *U* test (MW) was performed on unpaired contrasts of interest (ex: within cluster 1, survivor vs deceased). If multiple MW were performed in a same panel, we first performed a Kruskal-Wallis (KW) test, then the MW was corrected for multiple comparisons with Dunn's multiple comparison test. For the comparison of categorical values (demographics table, Table 1), we applied $Chi^2$ test. For comparisons between three paired values (measurement of cytokine+ S-specific T cell response), we performed a Friedman test, with correction using Dunn's multiple comparison test. Participants with missing data (for example, who did not enough cells to perform stimulation with all three peptide pools) were excluded from both the panels and statistical analysis.

In the setting of pie charts, permutation tests (10,000 permutations) were calculated using the SPICE software (https://niaid.github.io/spice/). All other statistical tests were performed with Prism v9.5.0 (GraphPad). Statistical tests were considered two-sided and $p < 0.05$ was considered significant (bolded in the panels).

**Two-stage bootstrap**. The two-stage bootstrap [sometimes called Hierarchical Bootstrap[54]] is a resampling method that accounts for intra-patient correlation in instances of repeated measures. For a sample containing $n$ patients, possibly having repeated measures, it first creates a *bootstrap sample*, by randomly sampling with replacement $n$ patients. This entails that patients may be sampled more than once, or even be missing from the bootstrap sample. Then, for a given patient in the bootstrap sample, the second stage consists in randomly sampling its observations with replacement, generating a patient sub-sample with the same number of observations the patient had in the original sample. Again, in this stage, some of the patient's observations can be sampled multiple times, or missing, owing to the fact that sampling is done with replacement. All such patient sub-samples are aggregated into a full bootstrap sample. Typically, one generates a large amount of bootstrap samples and uses the distribution of the test statistic across these bootstrap samples as an estimate of the true underlying distribution, which can be hard, or even impossible, to derive analytically[55]. This method was used to compare, among the four patient clusters of the discovery cohort, (i) antibody kinetics; (ii) AUC for vRNA, and (iii) PHATE cytokine trajectories.

**Model of RBD-specific IgG, IgM, and IgA kinetics**. Kinetics of IgG, IgA and IgM antibody production were modeled using a logistic curve fit of log(antibody) ~ DSO with the *drm* function of the *drc* R package[56] set to the L.4 function (i.e. the 4-parameter logistic curve). While the lower limit of detection was that of the assay, we did not set any upper limit of detection; yielding in effect 3 parameters of estimation (the location parameter, the upper limit of detection, and the slope of the curve at the location parameter value). The pairwise differences between location parameters of these curves, i.e., the DSO at which 50% of maximal antibody production was reached, were used to compare clusters' antibody production kinetics. 1000 two-stage bootstrap samples were used to obtain confidence intervals around the estimates of pairwise differences between location parameters of the logistic curves. The main strength of this testing approach is that we need not rely on strong assumptions to find a suitable distribution for the test statistic (i.e. the pairwise difference between the DSO at which 50% of maximal antibody production was reached), thanks to bootstrapping. As such, this method is robust to misspecification of the kinetics model.

**Area under curve (AUC) on plasma viral RNA quantities over time.** To quantify viral load among viremic patients, we estimate average Viral Load (copies/ml) * Time (DSO) as an Area Under Curve (AUC) for each cluster, the curves being Generalized Additive Models (GAM) with smooth spline estimations of the relationship Viral Load ~ Time over the period ranging from DSO0 to DSO30. We used the Lower Level of Quantification (LLOQ) threshold of 65 copies/ml. We chose to not transform Viral Load to allow for interpretation of the AUC based on the original units (copies/mL * time). We used the R package mgcv function gam. Four[4] knots and gaussian kernel smoothing gave the best bias-variance tradeoff. AUC was computed for DSO0 to DSO25 to eliminate inherent instability at border (DSO30) of smooth spline fits. The fitted values at DSO0 were all equal to the LLOQ threshold, so no such instability was present. AUC inter-cluster pairwise differences were computed for 1000 two-stage bootstrap simulations.

To only consider patients with RNAemia, patient observations retained were those with at least 1 measurement over the LLOQ threshold within the DSO0-30 timeframe (Supplementary Table 2). The results were robust to the presence of outlier patient 268 from CHUM.

**Power calculation for the number of samples to test for RBD-specific B cells.** We hypothesized that the frequency of circulating RBD-specific B cells would be different in the low-antibody clusters compared to the high-antibody clusters (Cluster 2 vs Cluster 1 and Cluster 4 vs Cluster 3). To assess this hypothesis, we computed power curve estimates of RBD-specific B-cell counts (RBD.B) with appropriate power. As no useful previous estimates of effect sizes were found in existing literature, we used RBD-specific IgG relative light unit (RLU) (RBD.IgG) as a proxy of RBD-specific B cells (RBD.B), based on previous evidence of their association[29]. Relationship between RBD.B and RBD.IgG was determined based on a linear regression fit on a sample of $n = 14$ patients for which both measures were available; variability in this relationship was simulated using 1000 vanilla bootstrap simulations of the above sample. These 1000 regressions all at once were used to predict the values of RBD.B from RBD.IgG for 1000 distinct bootstrap samples of $n = 216$ patients during the acute phase of the infection (~DSO11). In each of these simulated samples, an effect size measure (Cohen's d) was computed for RBD.B. We thus obtain a distribution of effect sizes for both inter-cluster differences, from which we computed proper sample size according to a conservative estimate based on 97.5th percentile of the effect size distributions and a standard power of 0.8. The sample sizes obtained were $n = 32$ and $n = 9$ per cluster for inter-cluster differences Cluster 2 vs Cluster 1 and Cluster 4 vs Cluster 3, respectively.

Statistical models were generated using the following R packages: drc (v3.0-1) and mgcv (v1.9-1).

## Bulk RNA sequencing

**Sample collection, processing and sequencing.** For the discovery cohort, we utilized RNA sequencing data from the "core assays" of the BQC-19 Biobank. For information about the data access procedure, refer to https://en.quebeccovidbiobank.ca/analyses-de-bases-bqc19 and the "Data Availability" section. The technical procedures used to generate the BQC-19 transcriptomic data were as follows. Blood was collected into PAXgene Blood RNA tubes (BD Biosciences; San Jose, CA, USA) to ensure stabilization of intracellular RNA. Immediately after collection, tubes were inverted 10 times, kept at RT for 24 h, −20 °C for an additional 24 h, and stored at −80 °C. Batches of tubes were thawed overnight and total RNA was manually extracted using PAXgene Blood RNA Kit (Qiagen; Germantown, MD, USA), as per manufacturer instructions. Total RNA was quantified, and its integrity assessed on a LabChip GXII (PerkinElmer) instrument. Libraries were generated from 250 ng of total RNA as follows: mRNA enrichment was performed using the NEBNext Poly(A) Magnetic Isolation Module (New England BioLabs). cDNA synthesis was achieved with the NEBNext RNA First Strand Synthesis and NEBNext Ultra Directional RNA Second Strand Synthesis Modules (New England BioLabs). The remaining steps of library preparation were done using the NEBNext Ultra II DNA Library Prep Kit for Illumina (New England BioLabs). Adapters and PCR primers were purchased from New England BioLabs. Libraries were quantified using the Kapa Illumina GA with Revised Primers-SYBR Fast Universal kit (Kapa Biosystems). Average size fragment was determined using a LabChip GXII (PerkinElmer) instrument. The libraries were normalized and pooled and then denatured in 0.05 N NaOH and neutralized using the HT1 buffer. The pool was loaded at 225 pM on an Illumina NovaSeq S4 lane as per the manufacturer's recommendations. The run was performed for 2 × 100 cycles (paired-end mode). A phiX library was used as a control and mixed with libraries at 1% level. Base calling was performed with RTA v3.4.4. Program bcl2fastq2 v2.20 was then used to demultiplex samples and generate fastq reads. The average base quality score for each sample dataset was verified to be Q33 or above and the percentage of aligned reads on reference sequence homo sapiens:hg19 was verified to be 90% or above.

For the validation cohort, we used the bulk RNA sequencing data that had been generated on PAXgene® Blood RNA tubes (BD, Franklin Lakes, NJ) prospectively collected in the Pronmed cohort ($n = 32$). These samples were distributed across clusters V1 ($n = 18$) and V2 ($n = 14$). Samples were allowed to stabilize at room temperature and

then frozen at −80 °C until used for RNA isolation. RNA extraction was done with the PAXgene blood RNA kit (Qiagen, Hilden, Germany, product no: 762174) according to manufacturer's instructions. RNA was quantified using a nanoDrop spectrophotometer (Thermo Fisher Scientific, Waltham, MA, USA) and quality measured using a BioAnalyser (Agilent, Santa Clara, CA, USA). Sequencing libraries were prepared from 400 ng (100 ng for two samples) total RNA using the TruSeq stranded mRNA library preparation kit (cat# 20020595, Illumina Inc. San Diego, CA, USA) including polyA selection. Unique dual indexes (cat# 20022371, Illumina Inc.) were used. The library preparation was performed according to the manufacturer's protocol (#1000000040498). Sequencing was performed as paired-end 150 bp read length on a NovaSeq 6000 system, S4 flowcell using v1.5 sequencing chemistry.

**Data processing and quality control.** For the discovery cohort, the sequencing reads were trimmed using CutAdapt[57] and mapped to the human reference genome (hg19) using 2STAR[58] aligner (version 2.6.1d), with default parameters. In the BQC19 transcriptomics dataset, only samples containing bulk RNA-sequencing data and assigned to PHATE clusters were retained for downstream analysis. (n = 445). Expression data was filtered for protein-coding genes that were sufficiently expressed across all samples (median logCPM > 1, n = 10,236 genes retained after filtering). After removing non-coding and lowly-expressed genes, normalization factors to scale the raw library sizes were calculated using calcNormFactors in edgeR (v3.26.8)[47]. The voom[59] function in limma (v3.40.6) was used to apply these size factors, estimate the mean-variance relationship, and convert counts to logCPM values. The technical effect of collection center (i.e., Centre Hospitalier de l'Université de Montréal vs Jewish General Hospital) was regressed using limma (v3.40.6) prior to downstream modeling.

For the validation cohort, cutAdapt was used for trimming and STAR for sequence alignment. Feature quantification and classification was done using Salmon. Scaled merged gene counts where all samples had a gene count >0 from Salmon were quantile normalised and log2-transformed and used for analysis of differentially expressed genes between cluster 1 and 2 in the Pronmed dataset using a linear model (limma version 3.54.2 running under R version 4.3.2) without covariates since n = 32 was deemed too low for adjustment.

**Modeling PHATE effects.** For the discovery cohort, PHATE effects (i.e., the differential expression effects between individuals in PHATE clusters of interest) were modeled in individuals collected at the DSO11 timepoint with expression data (n = 174; n PHATE cluster 1 = 37, n PHATE cluster 2 = 35, n PHATE cluster 3 = 41, n PHATE cluster 4 = 61 individuals). All pairwise PHATE clusters contrasts were performed with the main contrasts of interest being cluster 1 vs 2, cluster 3 vs 4, and, within cluster 1 individuals, survivor (n = 19) vs deceased (n = 18). To obtain estimates of the PHATE effects, the following linear model was run for each pairwise PHATE contrast:

$$
\begin{aligned}
M1: E(i,j) \sim {} & \beta_0(i) + \beta_{PHATE}(i) \boxtimes PHATE(j) + \beta_{age}(i) \boxtimes age(j) \\
& + \beta_{BMI}(i) \boxtimes BMI(j) + \beta_{sex}(i) \boxtimes sex(j) \\
& + \beta_{flowcell}(i) \boxtimes flowcell(j) + \beta_{CBC1}(i) \boxtimes CBC1(j) \\
& + \beta_{CBC2}(i) \boxtimes CBC2(j) + \varepsilon(i,j)
\end{aligned}
$$

Here, $\beta_0(i)$ is the global intercept accounting for the expected collection center-corrected expression of gene i in a female individual in the baseline PHATE cluster, and $\beta_{PHATE}(i)$ indicates the effect of the non-baseline PHATE cluster (PHATE(j)) on gene i. For example, in the PHATE cluster contrast 1 vs 2, individuals in PHATE cluster 2 represent the baseline gene expression signature, and $\beta_{PHATE}(i)$ represents the effect of PHATE cluster 1 on gene expression. Further, age represents

the mean-centered, scaled (mean = 0, sd = 1) age per individual, body mass index (BMI) represents the mean-centered, scaled (mean = 0, sd = 1) BMI per individual, sex represents the assigned sex for each individual (factor levels = "Female", "Male"), and flow cell represents the flow cell on which the sample was sequenced (seven factors in total). If BMI was not reported for an individual, this missing data was filled with the average BMI value across all individuals. Because we modeled whole blood expression data, two additional covariates were included, corresponding to the first two principal components of a PCA performed on an n x m cell type proportion matrix (where n = number of samples = 630, m = number of cell types = 5, with the matrix populated by the cell type proportions derived from clinical complete blood count [CBC] data) to account for the majority of the variance introduced by underlying cell type composition (PC1 percent variance explained (PVE) = 88.3%, PC2 PVE = 7.8%, total = 96.1%). Their corresponding effects on gene expression are represented by $\beta_{CBC1}(i)$ and $\beta_{CBC2}(i)$. Finally, $\varepsilon(i,j)$ represents the residuals for each gene i, individual j pair.

These models were fit using the lmFit and eBayes functions in limma[60], and the estimates of the PHATE effects $\beta_{PHATE}(i)$ were extracted across all genes along with their corresponding p-values. We controlled for false discovery rates (FDR) using an approach analogous to that of Storey and Tibshirani[61,62], which makes no explicit assumptions regarding the distribution of the null model but instead derives it empirically. To obtain a null, we performed 100 permutations, where PHATE cluster label was permuted across individuals.

**Calculation of ssGSEA scores.** To construct the IFN and COVID-19 severity score metrics, we calculated single-sample Gene Set Enrichment Analysis (ssGSEA) scores using the Gene Set Variation Analysis (GSVA) package in R (v1.32.0) with default parameters and method = "ssgsea"[63]. For the IFN ssGSEA score, the input genes were those belonging to the hallmark IFN gamma and alpha response pathways[64]. For the COVID-19 severity ssGSEA score, the input genes were those previously described to be positively associated with increased COVID-19 susceptibility in peripheral blood mononuclear cells from COVID-19+ patients[23].

**Gene set enrichment analyses.** Gene set enrichment analyses were performed using two independent methods, including fgsea (https://bioconductor.org/packages/release/bioc/html/fgsea.html) and ClueGO[25]. The enrichment program specifications and the data in which they were used to assess enrichments are described below:

The R package fgsea (v1.10.1) was used to perform gene set enrichment analysis for the cluster 1 vs 2 effects, cluster 3 vs 4 effects, and cluster 1 survivor vs deceased effects using the H hallmark gene sets[24]. T-statistics were obtained directly from the topTable function in limma[60]. The background set of genes were those sufficiently expressed (i.e. passed the lowly-expressed gene filter threshold) in the whole blood expression data. The t-statistics were then ranked, and these pre-ranked t-statistics were used to perform the enrichment using fgsea with the following parameters: minSize = 15, maxSize = 500, nperm = 100,000. Normalized Enrichment Scores (NES) and Benjamini-Hochberg adjusted p-values output by fgsea were collected for each analysis, which derives false discovery rates using the empirical p-value distribution of the data.

Additionally, we performed gene set enrichment analysis separately for genes upregulated in cluster 1 and cluster 3 individuals (i.e., the low antibody response clusters) relative to all other genes tested using ClueGO (v2.5.7)[25] in functional analysis mode. The target set of genes was the list of significantly upregulated genes in the cluster 1 or cluster 3 individuals (in the 1 vs 2 and 3 vs 4 contrasts, respectively) and the background set was the list of all genes tested. Specifically, we tested for the enrichment of GO terms related to biological processes (ontology source: GO_BiologicalProcess-EBI-UniProt-

GOA_04.09.2018_00h00) using the following parameters: visual style = Groups, default Network Specificity, GO Term Fusion = TRUE, min. GO Tree Interval level = 3, max. GO Tree Interval level = 8, min. number of genes = 3, min. percentage of genes = 4.0, statistical test used = Enrichment (right-sided hypergeometric test), *p*-value correction = Bonferroni step down. For the graphical representation of the enrichment analysis, ClueGO clustering functionality was used (kappa threshold score for considering or rejecting term-to-term links set to 0.4). Only pathways with an FDR < 0.05 were reported.

Scripts and processed data: https://github.com/herandolph/COVID-19_PHATE.

### Flow cytometry assessment of PBMCs

**Antibodies and reagents.** All antibodies are listed in Supplementary Tables 3, 4 and 5. Antibodies are monoclonal and raised in mice or rats. All antibodies were validated by manufacturer and titrated with biological and/or isotype controls. SARS-CoV-2 Spike receptor binding domain (RBD) recombinant protein was expressed in Freestyle 293 F cells and purified by nickel affinity columns, as directed by the manufacturer (Thermo Fisher Scientific). The RBD preparations were dialyzed against phosphate-buffered saline (PBS) and purity was assessed, by SDS-PAGE and Coomassie Blue staining. We generated B cell probes by conjugating recombinant RBD proteins with Alexa Fluor 488 dye or Alexa Fluor 594 dye (Thermo Fisher Scientific) according to the manufacturer's protocol.

**Detection of RBD-specific B cells.** Cryopreserved peripheral blood mononuclear cells (PBMCs) were thawed and rested in cell culture media (RPMI supplemented with 10% fetal bovine serum (FBS) and PenStrep – 50 U/ml of penicillin and 50 µg/mL of streptomycin) at 37 °C for 3 h at a density of $1 \times 10^7$ cells/ml in 24-well plates. Cells were collected, washed, and stained with LIVE/DEAD™ Fixable Aqua Dead Cell Stain Kit (20 mins, 4 °C; Thermofisher, #L34965). After washing, cells were stained with a cocktail of surface markers (30 mins, 4 °C; See panel in Supplementary Table 3, including RBD probes). Washed cells were then fixed with 2% paraformaldehyde (PFA) for 20 mins at RT, then washed and resuspended in PBS-2% FBS for flow acquisition on a 5-laser Symphony (BD). Analyses were performed using FlowJo (Treestar, V10).

**Activation-induced marker (AIM) assay on T cells.** Cryopreserved PBMCs were thawed and rested in cell culture media (RPMI supplemented with 10% Human AB serum and PenStrep – 50 U/mL of penicillin and 50 µg/mL of streptomycin) at 37 °C for 3 h at a density of 10 M/mL in 24-well plates. 15 min prior to stimulation, CD40 blocking antibody (clone HB14, Miltenyi, cat #: 130-094-133) was added to each well at 0.5 µg/ml, as well as antibodies staining CXCR5, CXCR3 and CCR6. Cells were either left unstimulated or stimulated with overlapping peptide pools of Spike (S1 + S2), at a final concentration of 0.5 µg/mL/peptide. Alternatively, 1 µg/ml of Staphylococcal Enterotoxin B (SEB, Toxin Technology) was used to stimulate the cells as a positive control. Cells were stimulated for 15 h, collected, washed, and stained with LIVE/DEAD™ Fixable Aqua Dead Cell Stain Kit (20 mins, 4 °C; Thermofisher, #L34965). After washing, cells were incubated with FcR block (10 mins, 4 °C; Miltenyi) then stained with a cocktail of surface markers (30 mins, 4 °C; See panel in Supplementary Table 4). Washed cells were then fixed with 2% PFA for 20 mins at RT, then washed and resuspended in PBS-2% FBS for flow acquisition on a 5-laser Symphony (BD).

**Intracellular cytokine staining (ICS) in Spike-specific T cells.** Cryopreserved peripheral blood mononuclear cells (PBMCs) were thawed and rested for 2 h in cell culture media. Cells were stimulated with overlapping peptide pools for SARS-CoV-2 spike (S), membrane (M) and nucleocapsid (NC) (0.5 µg/ml per peptide from JPT, Berlin,

Germany) for 6 h in the presence of anti-CD107a BV786 (BD Biosciences), Brefeldin A (BD Biosciences) and monensin-1 (BD Biosciences) at 37 °C and 5% $CO_2$. DMSO-treated cells served as negative control and SEB-treated cells as positive control. Cells were stained with LIVE/DEAD™ Fixable Aqua Dead Cell Stain Kit (20 mins, 4 °C; Thermofisher, #L34965) and surface markers (30 mins, 4 °C), followed by detection of intracellular markers using the IC Fixation/Permeabilization kit (Thermo Fisher) according to the manufacturer's protocol before acquisition at 5-laser Symphony (BD) (see Supplementary Table 5 for panel).

### Reporting summary

Further information on research design is available in the Nature Portfolio Reporting Summary linked to this article.

## Data availability

The plasma analyte measurements (relating to Figs. 1 and 2) and the frequency of CoV-2-specific immune cells (relating to Figs. 4 and 5) of the discovery cohort, following appropriate batch corrections and normalizations, can be found in Source_Data_File.xlsx. Given that Tables 1 and 2 describe the demographics of our cohorts, we have represented age using age brackets instead of exact values for ethical reasons (to protect participant identities). We accessed raw sequencing and transcriptomic data upon request from the "Biobanque Québécoise de la COVID-19 (BQC19)" (Quebec COVID-19 Biobank) data repository (info@bqc19.ca; website: https://en.quebeccovidbiobank.ca/analyses-de-bases-bqc19). For more information about the access procedure and the data access agreement for interested investigators, visit https://en.quebeccovidbiobank.ca. The anonymized dataset of gene expression data, after quality control and alignment (without any Personally Identifiable Information, or any information that could allow identification of individuals in the studies), is available on Zenodo (https://doi.org/10.5281/zenodo.6963452). Due to the high number of contrasts, we have kept the results of the RNA Seq analyses (Fig. 3) separate. These can be found in Supplementary Data 2 and 3, with meta data necessary for its analysis in Supplementary Data 1. Data from the Pronmed study is available from the SciLifeLab data repository after appropriate permissions and data access agreements (https://doi.org/10.17044/scilifelab.14229410). The human genome hg19 was used for alignement and is available at: https://www.ncbi.nlm.nih.gov/datasets/genome/GCF_000001405.13/. Source data are provided with this paper.

## Code availability

The code we used to perform the PHATE clusters analyses is available on Github at https://github.com/sachaMorin/covid-plasma-clusters. The DOI to the deposited notebook is https://doi.org/10.5281/zenodo.10912522. All code we used for analyzing the bulk RNA-Seq data can be found on GitHub at https://github.com/herandolph/COVID-19_PHATE.

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

## Acknowledgements

*Individuals and Facilities:* The authors are grateful to the study partici-pants and the clinical research teams. We thank the CRCHUM BSL3 and Flow Cytometry Platforms for technical assistance. The authors thank the Mila COVID-19 task force for fruitful discussions and feedback during the conception, development and application of the methods presented here. The Pronmed study acknowledges the clinical research team, particularly the research nurses Joanna Wessbergh and Elin Söderman and the biobank research assistants Erik Danielsson, Philip Karlsson, Labolina Spång and Amanda Svensson for their help in compiling the study. This study was funded by the American Foundation for AIDS Research (amfAR) grant 110068-68-RGCV (DEK, NC, AF); Canada's COVID-19 Immunity Task Force (CITF), in collaboration with the Canadian Institutes of Health Research (CIHR) grant VR2-173203 (DEK, AF); Canada CIFAR AI Chair (G.W.); NSERC Discovery grant 03267 (G.W.), NIH grant R01GM135929 (G.W.), CIHR grant # 178344 (DEK, AF); CIHR grants 365825 and 409511 (JBR); Canada Foundation for Innovation (CFI): Exceptional Fund COVID-19 grant #41027 to AF, DEK, NC; CFI leader to JBR. The Symphony flow cytometer was funded by a John R. Evans Leaders Fund from the Canada Foundation for Innovation (#37521 to D.E.K) and the Fondation Sclérodermie Québec. The Ministère de l'Économie et de l'Innovation du Québec, Programme de soutien aux organismes de recherche et d'innovation (AF), CRCHUM Foundation. The Biobanque Québécoise de la COVID-19 (BQC19) is supported by the Fonds de recherche Québec-Santé (FRQS) Génome Québec and the Public Health Agency of Canada. DEK and JBR are FRQS Merit Research Scholars. NC, MD, MC, JBR, CL and MT are supported by FRQS Salary Awards. AF and AP are recipients of Canada Research Chairs. EBR is recipient of a COVID-19 excellence scholarship from the Université de Montréal (EBR); SM is recipient of an IVADO MSc Excellence scholarship and a Fonds de recherche du Québec—Nature et technologies (FRQNT) B1X scholarship (SM); HER is supported by a National Institutes of Health (NIH) Ruth L. Kirschstein National Research Service Award (F31-HL156419); SPA, J.P., M.B. were supported by CIHR fellowships; G.S is supported by a FRQS doctoral fellowship and by a scholarship from the Department of Microbiology, Infectious Disease, and Immunology of the University of Montreal. The Pronmed study was funded by the SciLife-Lab/Knut and Alice Wallenberg national COVID-19 research program (M.H.: KAW 2020.0182, KAW 2020.0241), the Swedish Heart-Lung Foundation (M.H.: 20210089, 20190639, 20190637), the Swedish Research Council (R.F.: 2014-02569, 2014-07606), the Swedish Kidney Foundation (R.F.: F2020-0054) and the Swedish Society of Medicine (M.H.: SLS-938101). Funding bodies had no role in the design of the study, data collection, interpretation, or in the writing of the manuscript.

## Author contributions

Conceptualization: E.B.R., S.M., H.E.R., L.B., G.W., D.E.K. Data Curation: E.B.R., M. Labrecque, A. Pagliuzza, L.M., M.H., E.B., S.Z., T.N., D.M., J.R., C.L., A. Prat, N.A. Formal analysis: E.B.R., S.M., H.E.R., M. Labrecque, J.B., R.L.B., A.P., L.M., M.H., A.M. Funding acquisition: D.E.K., G.W., L.B., A.F., N.C., J.B.R., M.T., J.C., D.W. Investigation/experiments: E.B.R., A. Pagliuzza, L.M., M.H., J.N., R.C., A.S.F., M.B., J.P., S.D., S.P.A., G.S., H.Z., M. Lipcsey, R.F., A.L., D.V., F.P., D.W., C.B., G.G.L., H.M., Biostatistical methodology: J.B., R.L.B., H.R. Patient recruitment and cohort adminis-tration: N.B., D.M., C.L., A. Prat, J.L.C., N.A., M.D., J.B.R., D.E.K. Visuali-zation: E.B.R., S.M., H.E.R., J.B., R.L.B., G.S. Supervision: M.T., N.C., L.B., K.M., G.W., J.R., A.F., D.E.K. Writing—original draft: E.B.R., S.M., H.E.R., D.E.K. Writing—review and editing: all authors.

## Competing interests

J.B.R. has served as an advisor to GlaxoSmithKline and Deerfield Capital. T.N. has received speaking fee from Boehringer Ingelheim for talks unrelated to this research. D.E.K. has served as an advisor to AstraZe-neca. These agencies had no role in the design, implementation, or interpretation of this study. The authors declare that they have no other competing interests.

## Additional information

**Peer review information** *Nature Communications* thanks Pablo Pena-loza-MacMaster, and the other, anonymous, reviewers for their con-tribution to the peer review of this work. A peer review file is available.

Elsa Brunet-Ratnasingham[1,2,30,33], Sacha Morin[3,4,33], Haley E. Randolph [5,33], Marjorie Labrecque[1,6], Justin Bélair [7,31], Raphaël Lima-Barbosa[7,31], Amélie Pagliuzza[1], Lorie Marchitto[1,2], Michael Hultström [8,9,10,11 ✉], Julia Niessl [1,2,32], Rose Cloutier[1], Alina M. Sreng Flores[1], Nathalie Brassard [1], Mehdi Benlarbi[1,2], Jérémie Prévost[1,2], Shilei Ding[1], Sai Priya Anand[1,2], Gérémy Sannier [1,2], Amanda Marks [12], Dick Wågsäter[9], Eric Bareke[1], Hugo Zeberg [13,14], Miklos Lipcsey [8,15], Robert Frithiof [8], Anders Larsson [16], Sirui Zhou [11,17], Tomoko Nakanishi [11,17,18,19], David Morrison [11], Dani Vezina [1,2], Catherine Bourassa[1], Gabriella Gendron-Lepage[1], Halima Medjahed[1], Floriane Point[1], Jonathan Richard[1], Catherine Larochelle [1,20], Alexandre Prat [1,20], Janet L. Cunningham [21], Nathalie Arbour [1,20], Madeleine Durand[1,22], J. Brent Richards [10,11,17,23], Kevin Moon[24], Nicolas Chomont [1,2], Andrés Finzi[1,2,25], Martine Tétreault[1,20], Luis Barreiro [5,26,27], Guy Wolf [3,4,7 ✉] & Daniel E. Kaufmann [1,28,29 ✉]

[1]Centre de Recherche du Centre Hospitalier de l'Université de Montréal (CRCHUM), Montreal, QC, Canada. [2]Département de Microbiologie, Infectiologie et Immunologie, Université de Montréal, Montreal, QC, Canada. [3]Department of Computer Science and Operations Research, Université de Montréal, Montreal, QC, Canada. [4]Mila-Quebec AI Institute, Montreal, QC, Canada. [5]Committee on Genetics, Genomics, and Systems Biology, University of Chicago, Chicago, IL, USA. [6]Bioinformatics Program, Department of Biochemistry and Molecular Medicine, Université de Montréal, Montreal, QC, Canada. [7]Department of Mathematics and Statistics, Université de Montréal, Montreal, QC, Canada. [8]Anaesthesiology and Intensive Care Medicine, Department of Surgical Sciences, Uppsala University, Uppsala, Sweden. [9]Integrative Physiology, Department of Medical Cell Biology, Uppsala University, Uppsala, Sweden. [10]Department of Epidemiology, Biostatistics and Occupational Health, McGill University, Montreal, QC, Canada. [11]Lady Davis Institute for Medical Research, Jewish General Hospital, Montreal, QC, Canada. [12]Department of Immunology, Genetics and Pathology, Uppsala University, Uppsala, Sweden. [13]Department of Neuroscience, Karolinska Institutet, Stockholm, Sweden. [14]Department of Evolutionary Genetics, Max Planck Institute for Evolutionary Anthropology, Leipzig, Germany. [15]Hedenstierna Laboratory, Department of Surgical Sciences, Uppsala University, Uppsala, Sweden. [16]Clinical Chemistry, Department of Medical Sciences, Uppsala University, Uppsala, Sweden. [17]Department of Human Genetics, McGill University, Montreal, QC, Canada. [18]Kyoto-McGill International Collaborative School in Genomic Medicine, Gaduate School of Medicine, Kyoto University, Kyoto, Japan. [19]Research Fellow, Japan Society for the Promotion of Science, Tokyo, Japan. [20]Department of Neurosciences, Université de Montréal, Montreal, QC, Canada. [21]Department of Medical Sciences, Psychiatry, Uppsala University, Uppsala, Sweden. [22]Centre Hospitalier de l'Université de Montréal (CHUM), Montreal, QC, Canada. [23]Department of Twin Research, King's College London, London, UK. [24]Department of Mathematics and Statistics, Utah State University, Logan, UT, USA. [25]Department of Microbiology and Immunology, McGill University, Montreal, QC H3A 2B4, Canada. [26]Section of Genetic Medicine, Department of Medicine, University of Chicago, Chicago, IL, USA. [27]Committee on Immunology, University of Chicago, Chicago, IL, USA. [28]Département de Médecine, Université de Montréal, Montreal, QC, Canada. [29]Division of Infectious Diseases, Department of Medicine, Lausanne University Hospital and University of Lausanne, Lausanne, Switzerland. [30]Present address: Department of Medicine, University of California, San Francisco, CA, USA. [31]Present address: Independent Data Scientist, JB Consulting, Montreal, QC H3S1K8, Canada. [32]Present address: BioNTech SE, Mainz, Germany. [33]These authors contributed equally: Elsa Brunet-Ratnasingham, Sacha Morin, Haley E. Randolph. ✉e-mail: michael.hultstrom@mcb.uu.se; guy.wolf@umontreal.ca; daniel.kaufmann@chuv.ch

