## [Peer Review File · Nature Communications]

REVIEWER COMMENTS

Reviewer #1 (Remarks to the Author):

This is a comprehensive paper that profiled 782 longitudinal plasma samples from hospitalized COVID-19 patients. Their data are consistent with the “Interferon paradox” model previously described in the literature using other models, which showed that excessive IFN signaling could have a negative impact on adaptive immunity.

Although the findings are not completely novel, I think this is an important work as it extends prior findings with other viral models SARS-CoV-2. I don't have any negative comments, I guess the only concern is that they should cite a little better the prior work by others. For example, the authors should cite prior work by Palacio et al showing that IFN-I responses can impair adaptive immune responses. I wonder if the mechanism is similar to the one suggested by Palacio and others in that paper. In other words, excessive IFN (especially IFN-I) affects various steps of the viral life cycle, which reduces antigen availability, limiting the priming of adaptive immune responses, since adaptive immune responses are critically dependent on the availability of antigen (Signal 1):

<https://rupress.org/jem/article/217/12/e20191220/152035/Early-type-I-IFN-blockade-improves-the-efficacy-of>

But overall, I don't have additional experiments to request and I think it is an important piece of work worth of publication.

Reviewer #3 (Remarks to the Author):

This is a very elegant work by Brunet-Ratnasingham and colleagues. I enjoyed reading through the text and the elegant analysis provided by the authors.

This study proposed immunovirological plasma profile to differentiate COVID-19 patient endotypes, providing insights into the disease's severity and outcomes. Authors showed that not only the

severity of the disease varied among patients but also their innate, antibody, and cellular responses to the virus. The major finding of this paper is a validation of several previous reports about the role of excessive Interferon (IFN) signaling in COVID-19 severity. It was found to affect the differential kinetics of virus-specific antibodies, B cells, and CD4+ T cell responses. Early robust immunity coupled with low inflammation and plasma viremia was associated with moderate disease and good prognosis. However, sustained IFN signaling coupled with high viremia, severe inflammation, and poor virus-specific B cell and CD4+ T cell responses correlated with high-fatality cases. The study underlines the value of classifying patients based on their immune response profiles rather than just clinical severity, allowing for more accurate prognosis and potential therapeutic strategies. It also suggests potential therapeutic targets for mitigating the effects of COVID-19.

Although the authors conducted an elegant analysis and cross-validated their observations using two cohorts, the study is lacking novelty given the role of excessive Interferon (IFN) signaling in COVID-19 severity was reported before.

Questions/concerns:

- 1- It will be interesting to see the outcome of the proposed approach using both cohorts combined and split into training and test set in a more unbiased way?
- 2- Are there any correlations between the cytokines and the tissue damage markers?
- 3- How do you reconcile the upregulation or increase of coagulation pathways in the survivor's group? given that COVID-19 severity is associated with a coagulation cascade?
- 4- Is there any stratification by sex or age? How about the difference between males and females? Is the IFN delay associated with gender?
- 5- In regards to the persistent Interferon responses and its association with severity, are there specific interferon genes/markers that are more associated with severity, in contrast to the host antiviral responses that is protective?
- 6- Another clustering approach would be appreciated to validate the different clusters, given that K-mean is sensitive to the initial number of clusters, to initial Centroids selection and local optima? Maybe algorithms such Elbow, silhouette analysis will help in initial cluster selection?

Reviewer #4 (Remarks to the Author):

The manuscript by Elsa et al. profiled 782 longitudinal plasma samples from 318 COVID-19 patients and found four clusters in a discovery cohort and two high-fatality clusters in a validation cohort. Most interestingly, critical and non-critical clusters with delayed antibody responses exhibit sustained IFN signatures, suggesting excessive IFN signaling delaying development of adaptive virus-specific immunity. Following are the comments that need to be addressed by the authors:

Major comments:

- 1) It is unclear what distance k-means clustering was used. If the Euclidean distance of the 14 measurements (vRNA, cytokines, tissue damage markers) was used, have the measurements been centered or scaled since their values have very different scales and ranges. It is unclear why k=4 was chosen? Has the between variances and within variances curve been estimated to find the elbow point?
- 2) In the discovery cohort, the seven cytokine markers was used to generate a cytokine score. However, in the validation cohort, five cytokines was measured but only IL-6 was compared between the two clusters. Why not use the cytokine score based on the five cytokines?
- 3) In the differential analysis, two FDR values were generated, one was based on BH, the other was based on Storey (Table S2). It is confusing to have two FDR values.
- 4) Are RNAseq performed on the two high-fatality clusters in the validation cohort? The transcriptomics difference between these two clusters reproducing the sustained IFN signatures would serve a strong evidence to support what has been found in the discovery cohort.
- 5) GSEA on the hallmark genes not only found IFN signaling but also uncovered other pathways, such as IL6_JAK_STAT3 signaling. It is unclear why IFN signaling is chosen rather than other pathways. In addition, the hallmark gene sets includes very limited number of pathways. MsigDB consists of other databases, like C7 (the immunologic signature gene sets), which might be more useful than hallmark.

Minor comments:

- 1) Page 11, $|\log_{FC}| > 0.05$ should be $|\log_{FC}| > 0.5$?
- 2) Enrichment scores (ES) in the GSEA were not comparable across gene sets. It is better to report normalized enrichment score (NES) rather than ES.
- 3) Figure 3. The y-axis of the volcano plots should be $-\log_{10}$ FDR instead of $-\log_{10}$ (pvalues).

Response to reviews. Manuscript NCOMMS-23-23435A “Sustained IFN signaling is associated with delayed development of SARS-CoV-2-specific immunity”

We thank the reviewers for the insightful comments. We believe that our manuscript has been significantly improved by incorporating their suggestions. To address all the concerns raised, we have made changes to the main figures, to the supplementary figures, and added eight “Figures for Reviewers” numbered R1-R8 in our response below.

Sincerely,

Daniel E. Kaufmann, M.D

Editorial and Reviewers' comments are in black, our responses in *blue italic*. Changes made to the text to address the reviews are **highlighted in yellow**.

Reviewer #1 (Remarks to the Author):

This is a comprehensive paper that profiled 782 longitudinal plasma samples from hospitalized COVID-19 patients. Their data are consistent with the “Interferon paradox” model previously described in the literature using other models, which showed that excessive IFN signaling could have a negative impact on adaptive immunity.

Although the findings are not completely novel, I think this is an important work as it extends prior findings with other viral models SARS-CoV-2. I don't have any negative comments, I guess the only concern is that they should cite a little better the prior work by others. For example, the authors should cite prior work by Palacio et al showing that IFN-I responses can impair adaptive immune responses. I wonder if the mechanism is similar to the one suggested by Palacio and others in that paper. In other words, excessive IFN (especially IFN-I) affects various steps of the viral life cycle, which reduces antigen availability, limiting the priming of adaptive immune responses, since adaptive immune responses are critically dependent on the availability of antigen (Signal 1):

But overall, I don't have additional experiments to request and I think it is an important piece of work worth of publication.

We thank the reviewer for their positive review of our work and the comment. We now cite and discuss the study by Palacio et al. in the context of our results. In this study, the authors experimentally infected mice with acute viral infections with concomitant early transient blockade of interferon signaling. One of their observations was an increase in viral titers. From this observation they suggest that increased viral antigen allowed for increased antigen presentation and it's part of the reason why there is an increase in antiviral specific adaptive immune responses. Our work only partially aligns with this observation: sustained interferon signaling was

observed in both the high virus cluster one as well as with the low virus cluster 3. We also observed only weak associations between the interferon score and the viral load in Supplemental Figure 3. It is important to note that's Palacio et al. also saw increased transcriptomic signatures of interferon even when interferon blockade, which is a more similar setting to our observations. Another possible explanation for these differences is that we exclusively looked in blood, while their work was done primarily in tissue.

A second observation by Palacio et al. was that early blockade of interferon signaling increased antigen presentation efficiency by antigen presenting cells, both through increased expression of MHC as well as increased antigen load. They then relate this increased antigen presentation to the boosted antiviral responses. These observations suggest that decreased interferon signaling does help antigen presenting cells present more efficiently and thus help the antiviral response. In line with these observations, our bulk RNA-Seq data has revealed a negative association between genes associated with MHC class II antigen presentation and processing such as CD74, CD86, HLA-DRB, with severity of the disease. These observations have been added to Figure S3 in response to this comment and to comment #5 of reviewer #3 (see Figure R4). Furthermore, we have added this citation and a sentence about this in our discussion.

Reviewer #3 (Remarks to the Author):

This is a very elegant work by Brunet-Ratnasingham and colleagues. I enjoyed reading through the text and the elegant analysis provided by the authors.

This study proposed immunovirological plasma profile to differentiate COVID-19 patient endotypes, providing insights into the disease's severity and outcomes. Authors showed that not only the severity of the disease varied among patients but also their innate, antibody, and cellular responses to the virus. The major finding of this paper is a validation of several previous reports about the role of excessive Interferon (IFN) signaling in COVID-19 severity. It was found to affect the differential kinetics of virus-specific antibodies, B cells, and CD4+ T cell responses. Early robust immunity coupled with low inflammation and plasma viremia was associated with moderate disease and good prognosis. However, sustained IFN signaling coupled with high viremia, severe inflammation, and poor virus-specific B cell and CD4+ T cell responses correlated with high-fatality cases. The study underlines the value of classifying patients based on their immune response profiles rather than just clinical severity, allowing for more accurate prognosis and potential therapeutic strategies. It also suggests potential therapeutic targets for mitigating the effects of COVID-19.

Although the authors conducted an elegant analysis and cross-validated their observations using two cohorts, the study is lacking novelty given the role of excessive Interferon (IFN) signaling in COVID-19 severity was reported before.

Questions/concerns:

1- It will be interesting to see the outcome of the proposed approach using both cohorts combined and split into training and test set in a more unbiased way?

We pondered this option. However, after consideration and additional analyses, we concluded that in our case it would not be the best approach, this for the following reasons:

- *In predictive epidemiological modelling, validation in diverse populations is considered more robust. Internal validation using random splits of a single cohort is prone to overestimating the accuracy of the model. A more complete argument is made by Frank Harrell in "Regression modelling strategies", second edition. Kapitel 5.3.1, pp 109-110.*
- *While the relationships between the different dimensions of data are reproduced in the validation cohort, it is important to note that the technical platforms used at both sites (Canada and Sweden) are different. There are therefore concerns regarding the mixing of data generated on both cohorts.*
- *However, we still wanted to test if it would be possible to do so with good results. We first attempted to combine both cohorts and test whether the modeling made sense on the whole data, before splitting the data into a training and test set (see Figure R1). In line with the differences in the platforms used to collect these measurements between the discovery and validation cohorts, there were notable differences in the distributions of the input variables (Figure R1A). As a result of these differences, the discovery and validation cohorts grouped in completely separate clusters, with the validation cohort forming its own separate cluster (Figure R1BC). As a consequence, the association between the clustering and outcome (Fig R1D) or severity (Fig R1E) are lost.*
- *We next tried to apply a batch correction by the center of origin (Fig R1F). Although this correction did fix the single-center cluster (Fig R1G – as all four clusters have datapoints from all three centers), there was still an absence of a high-fatality cluster (Fig R1H) and critical severity was distributed across all four clusters (Fig R1I). This was likely because differences in scales nulled the associations with fatal outcome. For example, the validation Pronmed cohort had no 'low vRNA' sample, as per the standards of the discovery cohorts. Thus, non-fatal outcomes would be computed as mid-vRNA levels, whereas non-fatal outcomes are low vRNA in the discovery cohorts. As this is repeated across most input variables, the associations between the variables and outcomes are lost.*

Figure R1. Merging discovery and validation cohorts is not possible due to differences in distributions of input variables. A) Comparisons of the distribution of input variables in common between the discovery and validation group, across all three cohorts: CHUM, JGH (discovery) and Pronmed (validation). **B-E)** PHATE embedding as a result of 0-centered and scaled (based on dataset statistics) on common variables, without batch correction. Embedding color-coded for **B)** k-means clustering ($k=4$); **C)** cohort origin; **D)** outcome within 60 days of symptom onset, where fatal is red, and survival is blue; **E)** severity of disease throughout hospitalization for COVID-19, where red is patients of critical severity, as characterized by the need for mechanical respiratory support, and blue are patients which did not require mechanical ventilation. **F-I)** PHATE embedding as a result of 0-centered and scaled (based on dataset statistics) on common variables, with batch correction for the center of origin applied. Embedding color-coded for **F)** k-means clustering ($k=4$); **G)** cohort origin; **H)** fatal (red) or non-fatal (blue) outcome within 60 days of symptom onset; **I)** critical (red) and non-critical (blue) severity.

2- Are there any correlations between the cytokines and the tissue damage markers?

To address this point, we performed Spearman correlations on the seven cytokine and three tissue damage markers detected across all patients of the discovery cohort at DSO11 (Fig R2A). We

observed that all 7 cytokines correlated positively with angiotensin-2 and RAGE. However, only CCL2, CXCL13 and TNF α correlated weakly with surfactant protein. This suggests tissue damage markers may have different relationships to inflammatory cytokines in the plasma of hospitalized COVID-19 patients. The overall trend, though, is a positive association between both groups of markers as demonstrated by the strong positive correlation between the cytokine score and the tissue damage score (Fig R2B). Associations between a more extensive set of cytokines and tissue damage markers, their link to severity in acute COVID-19, have been discussed in greater detail in our previous publication (Brunet-Ratnasingham et al., *Sci Adv* 2021). These associations have been added to Fig S1, and are discussed in the results section.

Figure R2. Association between cytokines and tissue damage markers. A) Correlation matrix of cytokines and tissue damage markers as measured in all patients of the discovery cohort at DSO11. B) Correlation between Cytoscore calculated on the seven cytokines, with the tissue damage score, calculated on the three tissue damage markers. Statistics = Spearman R. * $p < 0.05$; ** $p < 0.01$; *** $p < 0.001$.

3- How do you reconcile the upregulation or increase of coagulation pathways in the survivor's group? given that COVID-19 severity is associated with a coagulation cascade?

We thank the reviewer for this interesting point. We believe that two main factors contribute to this apparent discrepancy:

-An important part of the explanation is likely that in our study we are only looking at transcriptional signatures in blood, while the bulk (actually almost all) of the production of the proteins involved in the coagulation cascade takes place in the liver. The contribution of circulating blood cells to components of the coagulation cascade is quite small. Therefore, the coagulation transcriptional signature we observe in whole blood bulk RNAseq is not representative of protein synthesis by hepatocytes.

-While we have not measured plasmatic complement proteins in our study, it is important to note that the clearest evidence of activation of the coagulation cascade in people with severe COVID is based on plasma protein measurements (complement proteins or degradation products resulting from complement activation). Previous studies obtained clear results with either

targeted protein concentration measurements (e.g, de Nooier et al, JID 2021), or broader proteomics profiling (e.g, Demichev et al, PLOS Digit Health 2022) in plasma.

4- Is there any stratification by sex or age? How about the difference between males and females? Is the IFN delay associated with gender?

As shown in Supplemental Figure 1F, there were no differences of age and sex across the 4 clusters. Furthermore, we did not see “regions” in the PHATE embedding that were one sex over another (Figure R3A), nor one age group over another (Fig R3B). These observations suggest nor sex nor age drive the clustering of the datapoints. We have added these observations to Fig S1. Of note, we used the biological sex in the analyses of the paper (GWAS was used both to define biological sex and to profile genetic ancestry – see Figure S1J). Self-reported gender was also retrieved from the electronic medical records. Biological sex and self-reported gender as recorded in the EMR) were congruent for all participants in the cohorts studied.

To see whether these factors impacted the IFN response, we next looked at the ssIFN score at DSO11, as it was at this timepoint that the delay was most evident in our data. We compared the score across the whole cohort in males versus females (Fig R3C) and saw no differences between males and females. We next compared the interferon score at DSO11 within each individual cluster between males and females. In this contrast, we saw no differences between sexes across all four clusters (Fig R3C).

We next looked at the association between the interferon score and DSO 11 with the age of the patient when they acquired COVID-19 (Fig R3D). There was no significant association between both those metrics at the level of the entire cohort, although we observed a trend for a negative association. Again, when we examined each cluster individually, there was no significant association for clusters 1 and 2. A weak negative association with borderline p value was observed 4 clusters 3 and 4. Overall these results suggest that the IFN score during COVID-19 is not strongly associated nor with age, nor with sex, and is not driving the clustering of our data.

Figure R3. No or minimal association between age or sex with clustering or IFN signalling. A) Distribution of datapoints from male (red) or female (blue) patients across the PHATE embedding. B) Color-coding of datapoints based on the age of the patient the sample was taken from, with yellow being older age, and dark blue, younger age. Scale is shown on the right. C) Comparison of ssIFN score between male and female subjects across the whole discovery cohort, and within individual clusters. Stats = Mann-Whitney test. D) Correlation between age and ssIFN score. Dots are color-coded with respective cluster. Stats = Spearman across the whole discovery cohort, and within individual clusters.

Still, because of reported differences in COVID-19 driven by age and sex, we included age and sex were included as covariates in the linear model to estimate the PHATE effects.

5- In regards to the persistent Interferon responses and its association with severity, are there specific interferon genes/markers that are more associated with severity, in contrast to the host antiviral responses that is protective?

Yes, the reviewer is correct. We do find that certain interferon-related genes – defined as those in the Hallmark “Interferon alpha response” and “Interferon gamma response” sets – are more associated with severity than others. Genes known to be involved in MHC class II antigen presentation and processing, such as CD74, CD86, HLA-DRB1, and genes coding for interleukin receptors that serve to inhibit proinflammatory cytokines, such as IL10RA and IL18BP, are more highly expressed in less severe patients (Fig. R4, top row), indicating a role for the repression of inflammation in those patients.

On the other hand, certain genes, including CD274, FCGR1A, SOCS3, TNFAIP6, and UPP1, display higher expression in critical patients compared to less severe patients (Fig. R4, bottom row). Most of these genes are involved in immune- or inflammation-related processes, suggesting a multifaceted role for inflammatory signaling in the context of SARS-CoV-2 infection and differential infection outcomes. These data are now discussed in the results and discussion, and shown in Figure S3 (new panels S3K and S3L).

Figure R4. Interferon-related genes significantly ($|$ Pearson's $r| > 0.3$, $p < 1 \times 10^{-4}$) associated with patient severity at the time of sampling. Top row: genes negatively correlated with severity ($r < 0$); bottom row: genes positively correlated with severity ($r > 0$).

6- Another clustering approach would be appreciated to validate the different clusters, given that K-mean is sensitive to the initial number of clusters, to initial Centroids selection and local optima? Maybe algorithms such Elbow, silhouette analysis will help in initial cluster selection? *We agree that local minima and the centroid initialization are known limitations of the K-Means algorithm (Figure R5A). Our code in fact runs 10 initializations and picks the best one to mitigate*

this. We added a clarification to that effect in Data dimensionality reduction and clustering section in Material and Methods.

We tested two additional clustering methods to validate our k-means clusters: Spectral Clustering (Figure R5B) and Gaussian Mixtures (Figure R5C). We compare the new clusters with the k-means one using the Adjusted Rand Score, which ranges from -0.5 to 1.0 with 1.0 indicating a perfect match. The Adjusted Rand Score obtained using spectral clustering was of 0.82, while the adjusted Rand score obtained with gaussian mixture was of 0.75. Thus, although there are slight variances with the association of certain “fringe” datapoints to specific clusters, the assignment of a given datapoint to a specific cluster is conserved across clustering methods, further highlighting the robustness of our using the plasma profile to identify endotypes in this setting. As for the number of clusters, we refer the reviewer to the first comment of Reviewer #4 (Figure R6) where we perform an Elbow curve analysis.

Figure R5. Comparison of different clustering approaches. Visualization of cluster annotations on the PHATE embedding following clustering using **A)** K-Means approach ($n_clusters = 4$; approached used in our paper ; **B)** Spectral clustering. The sklearn Spectral Clustering estimator was run with $n_clusters=4$, $affinity='nearest_neighbors'$ and otherwise default parameters; **C)** Gaussian Mixture. The sklearn Gaussian Mixture estimator was run with $n_components=4$, $covariance_type='spherical'$ and otherwise default parameters.

Reviewer #4 (Remarks to the Author):

The manuscript by Elsa et al. profiled 782 longitudinal plasma samples from 318 COVID-19 patients and found four clusters in a discovery cohort and two high-fatality clusters in a validation cohort. Most interestingly, critical and non-critical clusters with delayed antibody responses exhibit sustained IFN signatures, suggesting excessive IFN signaling delaying development of adaptive virus-specific immunity. Following are the comments that need to be addressed by the authors:

Major comments:

1) It is unclear what distance k-means clustering was used. If the Euclidean distance of the 14 measurements (vRNA, cytokines, tissue damage markers) was used, have the measurements been centered or scaled since their values have very different scales and ranges. It is unclear why k=4 was chosen? Has the between variances and within variances curve been estimated to find the elbow point?

We used Euclidean distance of k-means clustering, and the measurements have been centered and scaled. These parameters have been further clarified in the *Data dimensionality reduction and clustering* section in Material and Methods.

We thank the reviewer for the suggestion and agree that an elbow curve would be useful. We consider the plot for the within-cluster sum of squares (WCSS) (Figure R6A). For better visualization, we also included a plot to visualize the WCSS gain by going from k-1 to k (Figure R6B).

Figure R6. Assessments of the number of k-mean clusters. A) Plot of within-cluster sum of squares (WCSS). B) Plot of WCSS from k-1 to k.

While the initial plot does not show a crisp elbow, the WCSS starts to plateau in the k=4 to k=6 range, where the gains become much smaller. We chose k=4 as the most parsimonious model. Further considerations for k=4 included PHATE visualizations and alignment with outcomes. We also considered the size of the clusters to avoid clusters with too few samples for downstream analyses.

2) In the discovery cohort, the seven cytokine markers was used to generate a cytokine score. However, in the validation cohort, five cytokines was measured but only IL-6 was compared between the two clusters. Why not use the cytokine score based on the five cytokines?

In the discovery cohort, the Cytokine score was calculated using the average values measured in 50 healthy controls (see Brunet-Ratnasingham et al., Sci Adv 2021 for details). No such cohort of healthy controls was available for the validation cohort. Furthermore, the differences in the platforms used to collect these measurements between the discovery and validation cohorts resulted in notable differences in the distributions of the input variables; as such, the healthy controls of the discovery cohort could not be used to generate a cytokine score in the validation cohort.

A work around this was to calculate a Z score using the average values in the overall validation cohort. These observations show an elevate cytokine score (i.e. overall cytokine content) in the high-fatality cluster V1, compared to cluster V2 (See Figure R7). These results align with our observations within the discovery cohort. We have decided not to add this within the manuscript as not to create confusion with differences in methods, and how they affect the scale.

Figure R7. Alternative Cytokine Score calculated for the validation cohort. For the five cytokines common between the discovery and validation cohorts, the Z-score was calculated, then averaged for all cytokines per datapoint. This is an alternative to the cytokine score, as no healthy control cohort was available for the validation cohort.

3) In the differential analysis, two FDR values were generated, one was based on BH, the other was based on Storey (Table S2). It is confusing to have two FDR values.

We opted to perform an FDR correction using the Storey-Tibshirani method in addition to our initial Benjamini-Hochberg (BH) correction because the Storey method derives false discovery rates using the empirical p-value distribution of the data. To avoid confusion, we have removed

the BH-corrected values from Table S2 in favor of the Storey FDR values, although both are strongly correlated (avg. Pearson's $r > 0.999$ for FDRs from the main cluster comparisons). All significance cut-offs referenced in the paper correspond to the Storey-corrected FDR values. This has been clarified in the Gene set enrichment analyses in the Material and Methods section of the manuscript.

4) Are RNAseq performed on the two high-fatality clusters in the validation cohort? The transcriptomics difference between these two clusters reproducing the sustained IFN signatures would serve a strong evidence to support what has been found in the discovery cohort.

We thank the reviewer for this important question. To address it, we performed bulk RNA Sequencing on all available PAXgene® Blood RNA tubes (BD, Franklin Lakes, NJ) prospectively collected in the Pronmed cohort (n=32). These samples were distributed across clusters V1 (n=18) and V2 (n=14). Samples were allowed to stabilize at room temperature and then frozen at -80°C until used for RNA was isolation. RNA was using the PAXgene blood RNA kit (Qiagen, Hilden, Germany, product no: 762174) according to manufacturer's instructions. RNA was quantified using a nanoDrop spectrophotometer (Thermo Fisher Scientific, Waltham, MA, USA) and quality measured using a BioAnalyser (Agilent, Santa Clara, CA, USA).

Sequencing libraries were prepared from 400 ng (100 ng for sample VF-3363-82 and VF-3363-84) total RNA using the TruSeq stranded mRNA library preparation kit (cat# 20020595, Illumina Inc. San Diego, CA, USA) including polyA selection. Unique dual indexes (cat# 20022371, Illumina Inc.) were used. The library preparation was performed according to the manufacturer's protocol (#1000000040498). Sequencing was performed as paired-end 150bp read length on a NovaSeq 6000 system, S4 flowcell using v1.5 sequencing chemistry.

cutAdapt were used for trimming and STAR for sequence alignment. Feature quantification and classification was done using Salmon. Scaled merged gene counts where all samples had a gene count > 0 from Salmon were quantile normalised and log2-transformed and used for analysis of differentially expressed genes between cluster 1 and 2 in the Pronmed dataset using a linear model (limma version 3.54.2 running under R version 4.3.2) without covariates since n=32 was deemed too low for adjustment. The analysis produced 48 significantly differentially expressed genes (eBayes, FDR < 0.05 , $|\logFC| > 0.5$).

We then performed gene set enrichment analysis using the hallmark database gene sets to identify global pathways which are enriched in the high fatality cluster of validation cohort compared to its low fatality cluster. In this contrast the high fatality cluster was again most highly enriched in both interferon type I and type II response pathways (Fig R8). It was also enriched in other pathways highly expressed in the clusters with delayed antibody responses of the discovery cohort: TNFa signaling via NFkB, inflammatory response and IL6 JAK STAT3 signaling. These results validate the enrichment of IFN responses in the discovery cohort. These important validation studies have been added as the new panel S3J to Fig S3, and are discussed in the results.

Figure R8. Gene set enrichment analysis on differentially-expressed genes between cluster V1 and V2 in validation cohort, using the Hallmark gene sets. Only pathways with adjusted $p < 0.1$ shown. The size of the circle is representative of the adjusted p value, while the color gives the direction of the enrichment: red means enriched in cluster V1; blue is an enrichment in cluster V2. These results are now included in the revised manuscript as Figure S3J.

5) GSEA on the hallmark genes not only found IFN signaling but also uncovered other pathways, such as IL6_JAK_STAT3 signaling. It is unclear why IFN signaling is chosen rather than other pathways. In addition, the hallmark gene sets includes very limited number of pathways. MsigDB consists of other databases, like C7 (the immunologic signature gene sets), which might be more useful than hallmark.

We chose to focus on the interferon signaling pathways because their enrichments were by far the strongest among those tested for both main cluster comparisons (cluster 1vs2: IFN-gamma NES = 3.59, IFN-alpha NES = 3.77; cluster 3vs4: IFN-gamma NES = 3.49, IFN-alpha NES = 3.55). All other immune-related Hallmark pathways were not as strongly enriched ($|NES| < 2.7$).

Initially, we decided to focus on the Hallmark gene sets because, although they are relatively limited in number, they represent well-defined processes and are thus easy to interpret. While broader, the C7 sets are often much more difficult to interpret as they capture pathways defined in various immunological studies of different cell states, cell types, and perturbations. In contrast, the gene ontology (GO/C5) sets are more comprehensive than the Hallmark sets and are also generally more interpretable than the C7 sets. If we consider GO/C5 enrichment analysis for the main cluster contrasts, we find that interferon-related GO pathways, such as “Response to interferon-beta” and “Response to interferon-alpha”, are strongly enriched ($FDR < 0.01$, $\logFC > 0$) among upregulated genes (Figs. S3H and S3I), in line with our Hallmark enrichment results.

Minor comments:

1) Page 11, $|\logFC| > 0.05$ should be $|\logFC| > 0.5$?

Yes, the reviewer is correct– this has been corrected in the text.

2) Enrichment scores (ES) in the GSEA were not comparable across gene sets. It is better to report normalized enrichment score (NES) rather than ES.

We agree and thank the reviewer for the suggestion. We have updated the relevant figures (Figs. 3G and S3G) and Table S3 to reflect normalized enrichment score instead of enrichment score. We have also updated the Gene set enrichment analyses in the Material and Methods section of the manuscript

3) Figure 3. The y-axis of the volcano plots should be $-\log_{10} FDR$ instead of $-\log_{10}(p\text{values})$.

The y-axes of the volcano plots are correctly labeled (i.e., p-values are plotted). However, the dashed lines represent the nominal p-values corresponding to an $FDR = 0.05$ and points with an $FDR < 0.05$ are highlighted in color. This has been clarified in the legend of Figure 3.

REVIEWERS' COMMENTS

Reviewer #4 (Remarks to the Author):

The authors have addressed my comments.

Response to reviews. Manuscript NCOMMS-23-23435B “Sustained IFN signaling is associated with delayed development of SARS-CoV-2-specific immunity”

We thank the reviewers for the insightful comments on the initial version of the manuscript. We believe that our manuscript has been significantly improved by incorporating their suggestions. They did not raise any new concerns on the version NCOMMS-23-23435A of the paper.

Changes made in version NCOMMS-23-23435B to address editorial and reporting requirements are highlighted in yellow in the text.

Sincerely,

Daniel E. Kaufmann, M.D

REVIEWERS' COMMENTS

Reviewer #4 (Remarks to the Author):

The authors have addressed my comments.